# Oxygen-independent organic photosensitizer with ultralow-power NIR photoexcitation for tumor-specific photodynamic therapy

Yufu Tang[1], Yuanyuan Li[2], Bowen Li[1], Wentao Song[1], Guobin Qi[1], Jianwu Tian[1], Wei Huang [2], Quli Fan [2] ✉ & Bin Liu [1] ✉

Photodynamic therapy (PDT) is a promising cancer treatment but has limitations due to its dependence on oxygen and high-power-density photoexcitation. Here, we report polymer-based organic photosensitizers (PSs) through rational PS skeleton design and precise side-chain engineering to generate $\cdot O_2^-$ and $\cdot OH$ under oxygen-free conditions using ultralow-power 808 nm photoexcitation for tumor-specific photodynamic ablation. The designed organic PS skeletons can generate electron-hole pairs to sensitize $H_2O$ into $\cdot O_2^-$ and $\cdot OH$ under oxygen-free conditions with 808 nm photoexcitation, achieving NIR-photoexcited and oxygen-independent $\cdot O_2^-$ and $\cdot OH$ production. Further, compared with commonly used alkyl side chains, glycol oligomer as the PS side chain mitigates electron-hole recombination and offers more $H_2O$ molecules around the electron-hole pairs generated from the hydrophobic PS skeletons, which can yield 4-fold stronger $\cdot O_2^-$ and $\cdot OH$ production, thus allowing an ultralow-power photoexcitation to yield high PDT effect. Finally, the feasibility of developing activatable PSs for tumor-specific photodynamic therapy in female mice is further demonstrated under 808 nm irradiation with an ultralow-power of 15 mW cm$^{-2}$. The study not only provides further insights into the PDT mechanism but also offers a general design guideline to develop an oxygen-independent organic PS using ultralow-power NIR photoexcitation for tumor-specific PDT.

Traditional photodynamic therapy (PDT), which utilizes photosensitizers (PSs), light and tissue-endogenous oxygen ($O_2$) to produce reactive oxygen species (ROS)[1–10], is an important clinical cancer treatment modality due to its minimal invasiveness and low cost[11–19]. However, the wide clinical use of PDT is limited by the conflict between tumor-inherent hypoxia and the $O_2$-dependent ROS production mechanism and high-power-density photoexcitation[20–23].

So far, most PSs are operated on the $O_2$-dependent Type-II PDT mechanism through energy transfer between excited PSs and adjacent tumor-endogenous $O_2$ to yield the toxic singlet oxygen ($^1O_2$) (Fig. 1a). Less $O_2$-dependent Type-I PSs also have been developed to produce superoxide radical ($\cdot O_2^-$) or hydroxyl radical ($\cdot OH$) through electron transfer between the excited PSs and adjacent tumor-endogenous $O_2$ (Fig. 1a)[1,5,24–32]. Type-I PSs reduce the $O_2$ dependency through partial

[1]Department of Chemical and Biomolecular Engineering, National University of Singapore, Singapore, Singapore. [2]Key Laboratory for Organic Electronics and Information Displays and Jiangsu Key Laboratory for Biosensors, Jiangsu National Synergetic Innovation Center for Advanced Materials, Institute of Advanced Materials, Nanjing University of Posts and Telecommunications, Nanjing, China. ✉e-mail: iamqlfan@njupt.edu.cn; cheliub@nus.edu.sg

**Fig. 1 | Design, structures, and characterization of PSs. a** Illustrating Type-I electron transfer mechanism based on $O_2$-dependent $\cdot O_2^-$ and $\cdot OH$ production, Type-II energy transfer mechanism based on $O_2$-dependent $^1O_2$ production and our proposed mechanism based on photosensitized other substrates than $O_2$ for $O_2$-independent PDT. **b** The detailed molecule skeleton design strategy toward our proposed organic PSs with low bandgap and appropriate redox potential to sensitize $H_2O$ into $\cdot O_2^-$ and $\cdot OH$ for achieving 808 nm photoexcitation and $O_2$-independent $\cdot O_2^-$ and $\cdot OH$ production. **c** Schematic illustration of proposed detailed side-chain engineering strategy to boost $\cdot O_2^-$ and $\cdot OH$ production by using glycol side chains instead of commonly used alkyl side chains. Glycol side chains show

higher permittivity than alkyl side chains to inhibit rapid electron−hole recombination. In addition, glycol side chains display stronger hydrophilicity than alkyl side chains to bring more $H_2O$ molecules into the PS nanoparticles for participating in photosensitized reactions with electron−hole pairs. **d** Chemical structures of NTOalk, NTalk, and NTgly. **e** Schematic illustration of the preparation of PS nanoparticles through nanoprecipitation. **f** DLS profiles of PS NPs dispersed in PBS buffer (pH 7.4). Inset: the representative TEM photographs of NTgly NPs. **g** Normalized absorption spectra of PS NPs in water. **h** The fluorescence spectra of PS NPs in water. [NPs] =10 μg mL$^{-1}$.

oxygen circulation based on Haber−Weiss reactions or superoxide dismutase (SOD)-mediated disproportionation reactions (Supplementary Fig. 1), but still require tumor-endogenous $O_2$[1]. Recently, several innovative strategies have been developed to overcome hypoxia by delivering $O_2$ carriers or $O_2$-producing reagents, such as perfluorocarbons, artificial red blood cells, and metal oxide catalysts[6,20,22,23,33−36]. These designs suffer from several limitations, such as complicated compositions and spatiotemporal uncontrollability. To eliminate the dependence on tumor-endogenous $O_2$, PS was defined as a new generation of PDT that did not involve tumor-endogenous $O_2$ through energy or electron transfer between excited PSs and adjacent

other substrates, such as proteins, RNA, and redox species (Fig. 1a)[37,38]. So far, only a few such PSs were reported to be able to transfer photogenerated electron−hole ($e^-−h^+$) pairs into nearby intracellular redox substrates, such as nicotinamide adenine dinucleotide (NADH), GSH, and protein disulfide bonds[39−44]. However, the efficiency of these PSs can still improve so much more because these locally distributed, large-size and exhaustible redox substrates in cells may not be close to PSs for efficient photosensitized reactions. In addition, most electron−hole pairs need to successfully migrate to the PS nanoparticle's surface in order to participate in the photosensitive reaction with substances for these PSs. Unfortunately, the migration length of

electron–hole pairs is typically less than 20 nm[45]. Thus, these PSs generally required a relatively high-power photoexcitation (>100 mW cm$^{-2}$)[39,46], leading to undesirable side effects. Compared with the above substrates, $H_2O$ is abundant and ubiquitous in tissues, which is a more desirable substrate for PSs. Theoretically, holes with enough high oxidation potential (>0.81 V vs. normal hydrogen electrode (NHE), $2H_2O + 4 h^+ \rightarrow O_2 + 4H^+$, pH = 7[47,48]) can oxidize $H_2O$ into $O_2$ and $H^+$, while photogenerated electrons with sufficiently low reduction potential (<−0.33 V vs. NHE[48], $O_2 + e^- \rightarrow \cdot O_2^-$, pH = 7 or <0.31 V vs. NHE[49–52], $O_2 + 3H^+ + 3e^- \rightarrow \cdot OH + H_2O$, pH = 7) can be used to further reduce $O_2$ into $\cdot O_2^-$ or $\cdot OH$. Such an approach of sensitizing $H_2O$ into $\cdot O_2^-$ and $\cdot OH$ fundamentally enables $O_2$-independent PDT. However, the reaction is restricted by dynamic factors of rapid recombination of electron–hole pairs and higher thermodynamic barriers of oxidation potential ( >0.81 V for $H_2O$-to-$O_2$) than the existing oxidation reaction of intracellular substrates (e.g., ≥0.26 V for GSH-to-GSSG evolution and ≥0.32 V for NADH-to-NAD$^+$ evolution)[44,53–56]. In addition, integrating high oxidation and low reduction potential in a single PS often leads to a wide bandgap, which causes these PSs to be excited by short wavelength light with limited tissue penetration depth in PDT.

Here, we show a general strategy through rational PS skeleton design and precise side-chain engineering to develop organic PSs with $O_2$-independent $\cdot O_2^-$ and $\cdot OH$ production upon ultra-low-power-density 808 nm photoexcitation (Fig. 1a–e). As a proof-of-concept, we use naphthalenediimide with glycol or alkyl side chains and bithiophene to synthesize the PSs (Fig. 1d). Conjugated PS skeletons endow deep-tissue-permeating 808 nm photoexcitation and generate electron–hole pairs (Fig. 1b, d). Holes possessing enough high oxidation potential satisfy the thermodynamic condition required for a photosensitized reaction to oxidize $H_2O$ into $O_2$ and $H^+$, while photogenerated electrons further reduce $O_2$ and $H^+$ into $\cdot O_2^-$ and $\cdot OH$. The process that sensitizes $H_2O$ into $\cdot O_2^-$ and $\cdot OH$ fundamentally enables $O_2$-independent PDT. Furthermore, we optimize the dynamic factors to improve $\cdot O_2^-$ and $\cdot OH$ yields by side-chain engineering that uses glycol oligomers instead of commonly used alkyl groups as side chains of PS skeletons (Fig. 1c). Glycol side chains increase permittivity, which could inhibit rapid electron–hole recombination. In addition, hydrophilic glycol oligomers also bring more $H_2O$ molecules into the electron–hole pair vicinity for photosensitization (Fig. 1c, d). Together, these effects can boost the production efficiency of $\cdot O_2^-$ and $\cdot OH$ and reduce the power density of 808 nm excitation light. Finally, we demonstrate the feasibility of developing activatable PSs for in vivo tumor-specific photodynamic ablation under 808 nm irradiation at 15 mW cm$^{-2}$, which is far below the maximum permissible exposure of skin (0.33 W cm$^{-2}$, the American National Standard) at 808 nm. Therefore, this work develops a universal protocol to provide an $O_2$-independent organic PS with ultra-low-power-density NIR photoexcitation for tumor-specific PDT.

## Results

### Design, preparation, and characterization of our proposed PSs

In general, a redox reaction is controlled by thermodynamic and dynamic factors. The thermodynamic factors determine the possibility of a reaction to occur, and the dynamic factors decide the reaction efficiency[57]. The design of our proposed NIR-excited PSs faces three challenges: (1) NIR-excited organic PSs should have an oxidation potential higher than 0.81 V vs. NHE ($H_2O + 4 h^+ \rightarrow O_2 + 4H^+$) and a reduction potential lower than −0. 33 V vs. NHE ($O_2 + e^- \rightarrow \cdot O_2^-$) or 0.31 V vs. NHE ($O_2 + 3H^+ + 3e^- \rightarrow \cdot OH + H_2O$) to satisfy the thermodynamic condition of photosensitized reaction to occur. Compared to designing PSs with low reduction potential, realizing our proposed PSs with a high oxidation potential (> +0.81 V) to oxidize $H_2O$ into $O_2$ is more difficult. Until now, great efforts have been devoted to oxidizing $H_2O$ into $O_2$. However, existing materials are almost based on inorganic or metallic-based semiconductors[58,59], such as carbon nitride, titanium

dioxide and Bi/BiOx, facing challenges in complicated components and potential long-term biotoxicity in vivo. By contrast, organic materials are preferred due to their relatively good biocompatibility. Although researchers have been devoted to developing organic semiconductors for oxidizing $H_2O$ into $O_2$, almost all of them are based on UV–Vis (300−650 nm) and high-power-density photoexcitation[60,61], which is not suited for use under biological conditions[48,59]. Thus, organic materials oxidizing $H_2O$ into $O_2$ in vivo under deep-tissue-penetrating near-infrared (NIR, 700−900 nm) excitation with ultra-low-power density have not been achieved so far. (2) The second challenge is how to reduce rapid electron–hole recombination to yield enough electrons and holes for participating in the $H_2O$ redox reaction, which is critical to a high yield of $\cdot O_2^-$ or $\cdot OH$. The common strategy focuses on designing PS conjugation skeletons by introducing heavy metal atoms or special organic moieties to facilitate the intersystem crossing process to increase the lifetime of electrons in excited triplet states of PSs[62]. However, PS conjugation skeleton modification is complex and often changes the photophysical properties of organic semiconductors. Another popular approach is using substances, such as GSH and NADH, to scavenge holes to reduce the recombination of electron–hole pairs on time[54]. However, these exhaustible and locally distributed substrates in cells may not be close enough to PSs for photosensitized reactions to occur effectively. (3) Traditional strategy commonly uses alkyl groups as side chains of PS-conjugated skeletons. Because of the strong hydrophobicity of PS-conjugated skeletons and alkyl side chains, most substances, such as $H_2O$, GSH, and NADPH, are difficult to diffuse into PS nanoparticles[63]. Therefore, the PDT-dependent photosensitive reactions will mainly happen on the surface of the nanoparticles with $H_2O$, which requires electron–hole pairs to successfully migrate on the surface of the PS nanoparticles to participate in the photosensitive reaction. However, the migration length of electron–hole pairs is typically less than 20 nm in polymer films[64]. Consequently, many electron–hole pairs are not participated in the PDT-involved photosensitized reactions, leading to reduced $\cdot O_2^-$ or $\cdot OH$ production. Therefore, ensuring sufficient $H_2O$ molecules around electron–hole pairs in PS nanoparticles is the third challenge for high-efficiency $\cdot O_2^-$ or $\cdot OH$ production.

According to frontier molecular orbital theory, coupling electron donors and acceptors with low HOMO is the most effective strategy to obtain NIR-excitable PSs with high oxidation potential (Supplementary Fig. 2). From the existing electron donors and acceptors (Supplementary Fig. 3), we chose naphthalenediimide as the electron acceptor and bithiophene as the electron donor to yield a polymeric PS skeleton with a high oxidation potential and a low reduction potential, satisfying the thermodynamic condition. In addition, naphthalenediimide shows a high planar structure and large conjugation, which is favorable for long-wavelength NIR absorption, implementing deep-tissue-penetrating 808 nm excitation.

To inhibit rapid electron–hole recombination for highly effective $\cdot O_2^-$ and $\cdot OH$ generation, glycol side chains were used to replace the widely used alkyl chains attached to the polymeric PS skeletons. Compared with alkyl chains, the presence of C−O dipoles of glycol chains increases in relative permittivity, which could reduce the coulombic attraction between photogenerated electrons and holes to reduce their recombination and prolong their lifetimes[65,66]. More importantly, it does not significantly impact their optical properties, such as the absorption spectrum. In addition, the hydrophilic glycol side chains of PS skeletons could bring more $H_2O$ molecules into the interior of PS nanoparticles for radical production. Together, these effects optimize dynamic factors to improve reaction efficiency between $H_2O$ and electron–hole pairs, yielding effective PDT even upon ultra-low-power-density photoexcitation.

To demonstrate our design concept, three semiconducting polymer PSs (NTOalk, NTalk, and NTgly) were synthesized by Stille coupling using thiophene or 3,3′-dimethoxy-2,2′-bithiophene as an

electron donor and naphthalenediimide with the alkyl or glycol side chains as an electron acceptor, respectively (Fig. 1d and Supplementary Fig. 4). All the polymers were fully characterized by $^1$H nuclear magnetic resonance and gel permeation chromatography (Supplementary Figs. 5–15). The number of average molecular weights (Mn)/polydispersity indices (PDIs) of NTOalk, NTalk, and NTgly were 16.8 kDa/1.226, 16.2 kDa/1.292, and 15.9 kDa/1.215, respectively.

Furthermore, the three semiconducting polymer PSs were used to prepare PS nanoparticles (NPs, i.e., NTOalk NPs, NTalk NPs, and NTgly NPs) through nanoprecipitation in the presence of an amphiphilic copolymer of poly(styrene-co-maleic anhydride) (PSMA, average MW ~1700) (Fig. 1e). The carboxyl groups of PSMA on the surface of PS NPs endowed them with good biocompatibility and structure stability in water. Dynamic light scattering (DLS) shows that the NPs of NTOalk, NTalk, and NTgly have sizes of 37 nm, 38 nm and 33 nm (Fig. 1f), respectively, while transmission electron microscopy (TEM) images show their spherical morphologies (Fig. 1f, inset and Supplementary Figs. 16 and 17). Meanwhile, the three PS NPs also show good structure stability during storage in PBS buffer (pH 7.4) containing 10% FBS at 37 °C for 48 h (Supplementary Fig. 18). The absorption spectra of NTOalk, NTalk, and NTgly in dichloromethane have a broad absorption band from 500 to 900 nm with the absorption maxima at 780, 660, and 595 nm, respectively (Supplementary Fig. 19). By contrast, the absorption maxima of the NTOalk NPs, NTalk NPs, and NTgly NPs in water exhibit a prominent redshift in comparison with the counterpart polymers in dichloromethane (Fig. 1g). The NTOalk NPs, NTalk NPs, and NTgly NPs also show a broad absorption band from 500 to 900 nm with absorption peaks at 820, 710, and 745 nm, respectively (Fig. 1g). The mass absorption coefficients of NTOalk NPs, NTalk NPs, and NTgly NPs are 34.8, 12.2, and 14.9 cm$^{-1}$ mg$^{-1}$ mL at 808 nm, respectively. The PS NPs also exhibit broad NIR-II emission, which should benefit NIR-II fluorescence imaging-guided PDT (Fig. 1h).

## Mechanism and optimization of •O$_2^-$ and •OH generation

The photostability of the PS NPs was first evaluated under 808 nm excitation with a high-power density of 330 mW cm$^{-2}$ for 12 min (Supplementary Fig. 20). No obvious photobleaching was observed. Next, the in vitro •O$_2^-$ and •OH generation capability of three PS NPs was evaluated using •O$_2^-$ fluorescent probe (dihydrorhodamine123, DHR 123) and •OH fluorescent probe (4-hydroxyphenyl-fluorescein, HPF) in normal and hypoxic conditions under the irradiation of an 808 nm laser at 15 mW cm$^{-2}$ for 5 min (Supplementary Figs. 21–28 and Fig. 2a–d). DHR 123 could react with •O$_2^-$ to yield green fluorescence with a peak at 525 nm. HPF can react with •OH to produce fluorescein with an emission maximum of 515 nm under 490 nm excitation. As shown in Fig. 2a–d, both DHR 123 and HPF do not absorb at 808 nm; therefore, their fluorescence does not increase in normal and hypoxic conditions (hypoxic conditions established in a glovebox with <0.01 ppm O$_2$). Meanwhile, pure NTOalk NPs, NTalk NPs, and NTgly NPs without 808 nm laser irradiation also did not cause any increase in the fluorescence of DHR 123 and HPF in normal and hypoxic conditions. By contrast, the fluorescence intensities of DHR 123/HPF for NTOalk NPs, NTalk NPs and NTgly NPs were increased by 2.56/1.46-fold, 22.6/21.8-fold, and 38/36.33-fold in normal oxygen conditions, respectively, while 1.38/1.15-fold, 18.66/18.3-fold, and 34.67/34.33-fold were observed in hypoxic conditions under 808 nm laser irradiation for 5 min, respectively. Encouragingly, the NTgly NPs showed the strongest ability to produce •O$_2^-$ and •OH among the three PSs in both normal and hypoxic conditions. As compared to the well-known type-I PS methylene blue (MB) (Fig. 2d)[67], NTalk NPs and NTgly NPs also displayed a better ability to generate •O$_2^-$ and •OH. The F/F$_0$ of DHR 123 and HPF for NTgly NPs achieved up to 51.8-fold stronger •O$_2^-$ and •OH production than that of MB (Fig. 2c, d) under hypoxic conditions. However, the •O$_2^-$ and •OH production of NTOalk NPs exhibited a significant decrease in hypoxic conditions compared to normal

oxygen conditions, demonstrating that NTOalk NPs belonged to an external O$_2$-dependent type-I PS. Almost no difference was observed for NTalk NPs and NTgly NPs in generating •O$_2^-$ and •OH under both normal and hypoxic conditions, indicating that NTalk NPs and NTgly NPs were O$_2$-independent PSs. The electron spin resonance (ESR) analysis further supported the conclusion using 5,5-dimethyl-1-pyrroline-N-oxide (DMPO) as the •O$_2^-$ and •OH trapping agent under the irradiation of an 808 nm laser. The ESR spectrum of NTgly NPs displayed a typical resonance signal of •O$_2^-$ (red stars) and •OH (blue stars) in both normal (Fig. 2e) and hypoxic conditions (Fig. 2f). By contrast, no signals of •O$_2^-$ and •OH were observed for either the NTgly NPs alone or the 808-nm laser irradiation alone in normal (Fig. 2e) and hypoxic (Fig. 2f) conditions. Furthermore, the NTgly NPs can produce a high level of ROS even in the absence of O$_2$ (Supplementary Fig. 29), which is consistent with the above results (Fig. 2f).

To demonstrate our design mechanism, we first verified the thermodynamic conditions. The redox potentials of the PS polymers of NTOalk, NTalk and NTgly were measured by using cyclic voltammetry (Supplementary Figs. 30–32). The as-determined potentials (vs NHE) are plotted in Fig. 2g. The reduction potentials of NTOalk, NTalk, and NTgly were −0.72 V, −0.66 V, and −0.64 V, respectively, while the oxidation potentials were 0.59 V, 1.12 V, and 1.15 V, respectively. Among them, the oxidation potentials of NTalk (1.12 V) and NTgly (1.15 V) were higher than that of the potential for H$_2$O/O$_2$, H$^+$ (0.81 V vs NHE), which provided a strong driving force for oxidizing H$_2$O into O$_2$ and H$^+$, satisfying thermodynamic conditions for the reaction to occur. However, the oxidation potential of NTOalk was lower than that of H$_2$O/O$_2$, H$^+$, demonstrating that NTOalk could not oxidize H$_2$O into O$_2$. In addition, the reduction potentials of NTOalk, NTalk, and NTgly were −0.72 V, −0.66 V, and −0.64 V, respectively, which were lower than the potentials of O$_2$, H$^+$/•OH (0.31 V vs NHE, pH = 7) and O$_2$/•O$_2^-$ (−0.33 V vs NHE, pH = 7), indicating that three PSs could transfer electrons to O$_2$ to generate •O$_2^-$ and •OH, agreeing with the detection results of DHR 123 and HPF probes.

To further verify that the oxygen element of •O$_2^-$/•OH was derived from H$_2$O, H$_2$O containing H$_2^{18}$O (v:v = 1:1) was used as the mixed solvent to trap $^{18}$O$_2$ generation by the mass spectra. Since electrons could rapidly reduce the generated $^{18}$O$_2$, sodium ascorbate was added simultaneously as an electron sacrifice agent. As shown in Fig. 2h, the signal at $m/z$ = 36 corresponds to the $^{18}$O$_2$ product, providing strong evidence that H$_2$O can be effectively oxidized to O$_2$ by NTgly NPs. By contrast, when type-I PS NTOalk NPs were used to conduct the same experiment, only the signal at m/z = 32 was observed, corresponding to $^{16}$O$_2$ from the air (Fig. 2i). To further verify that the $^{18}$O$_2$ product could be reduced into the •O$_2^-$ or •OH, we used coumarin to capture the •$^{18}$OH in H$_2$O containing H$_2^{18}$O (v:v = 1:1)[68], and monitored the formation of 7-hydroxy ($^{18}$O)-chromen (Mw = 164.04). Only when NTgly NPs were used with H$_2$O containing H$_2^{18}$O, a signal at $m/z$ = 163.0729 could be observed (Fig. 2j), but not for pure water (Fig. 2k) or in the presence of NTOalk NPs (Fig. 2l). The isotopic mass spectra data indicate that our PSs were independent of tissue-endogenous O$_2$ in PDT.

Next, we further verified that glycol side chains could optimize dynamic factors by increasing the permittivity to inhibit rapid electron−hole recombination and bringing more H$_2$O molecules into the interior of PS nanoparticles to ensure sufficient H$_2$O molecules around electron−hole pairs. We first measured the permittivity of the NTalk and NTgly polymers in dry films (Fig. 3a). The results indicated that glycol chains increased the permittivity of NTgly (9.6) compared to alkyl side-chain-based NTalk (2.8). The increases in relative permittivity could be attributed to the presence of C−O dipoles on the glycol side chains, which reduced the coulombic attraction between photogenerated electrons and holes to mitigate their recombination and prolong their lifetimes[65,66]. Next, the fluorescence spectra of NTalk NPs and NTgly NPs were measured at the same concentration. As shown in

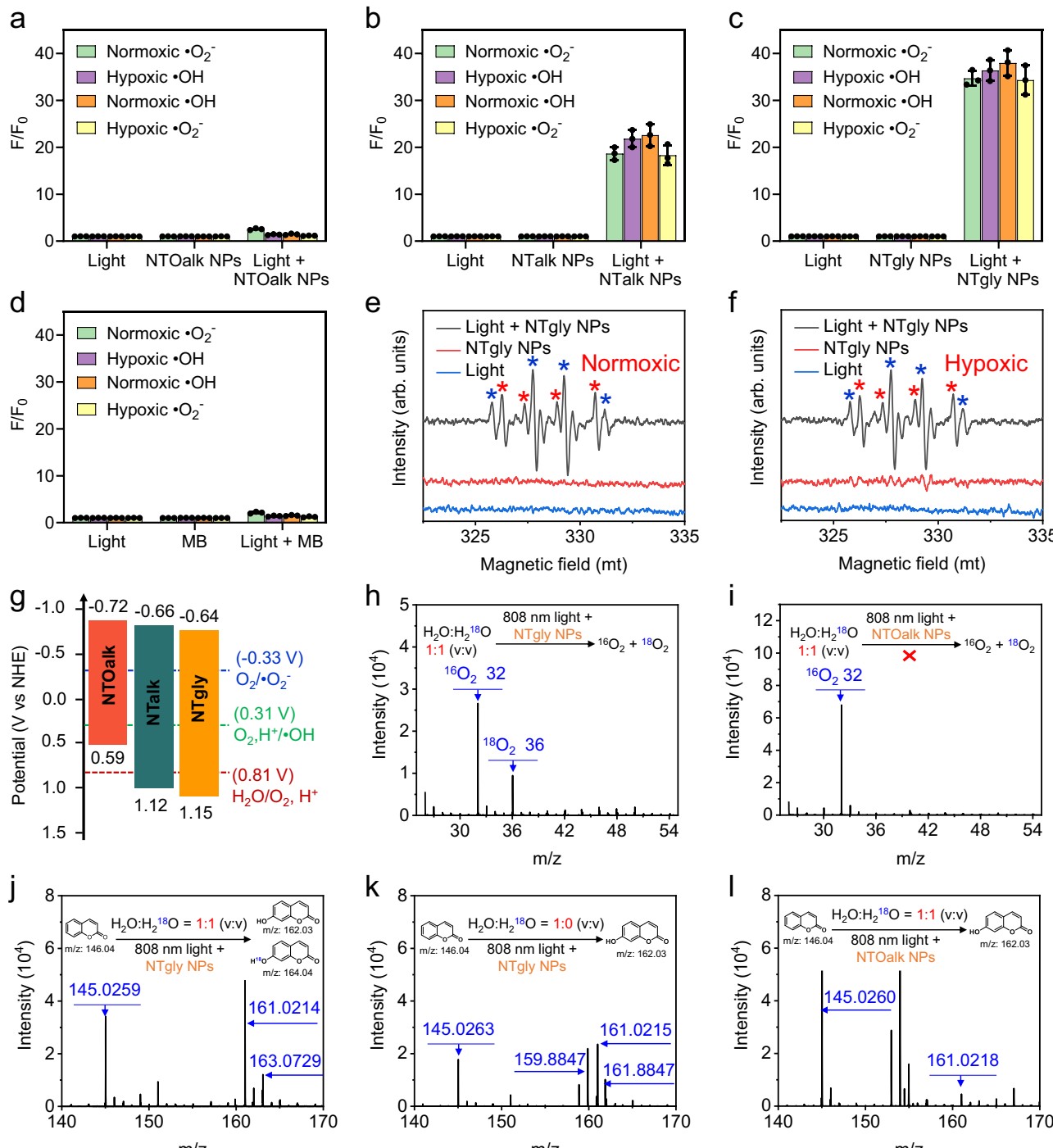

**Fig. 2 | Rational PS skeleton design for 808 nm laser-driven generation of •O₂⁻ and •OH under O₂-independent conditions.** The fluorescence intensity changes of •O₂⁻ probe (DHR 123) at 525 nm and •OH probe (HPF) at 515 nm for **a** NTOalk NPs, **b** NTalk NPs, **c** NTgly NPs, and **d** MB. (*n* = 3 independent samples; mean ± SD). Three PS NPs were 5 μg mL⁻¹ under 808 nm irradiation at 15 mW cm⁻² for 5 min. The typical resonance signal of •O₂⁻ (red stars) and •OH (blue stars) of ESR spectrum for NTgly NPs using DMPO as trapping agent under **e** normoxic and **f** hypoxic conditions. **g** Energy-level diagram of NTOalk, NTalk, and NTgly. Mass spectra of the products generated in ¹⁸O₂ using **h** NTgly NPs and **i** NTOalk NPs as the PSs in H₂O containing H₂¹⁸O (v:v = 1:1). Mass spectra of the products generated in the •OH with coumarin reaction using NTgly NPs as the PSs in **j** H₂O containing H₂¹⁸O (v:v = 1:1) and **k** pure H₂O. **l** Mass spectra of the products generated in the •OH with coumarin reaction using NTOalk NPs as the PSs in H₂O containing H₂¹⁸O (v:v = 1:1).

Fig. 3b, NTgly NPs showed lower fluorescence than NTalk NPs, indicating that glycol side chains strongly reduced radiative exciton recombination. Furthermore, we studied the photogenerated electron−hole pairs of NTalk and NTgly in films by monitoring the photocurrent density. As expected, the photocurrent density of NTgly (2.7 μA cm⁻²) was much higher than that of NTalk (0.75 μA cm⁻²)

(Fig. 3c), demonstrating that NTgly produced more photogenerated electron−hole pairs. The permittivity, fluorescence and photocurrent results indicated that glycol side chains could inhibit electron−hole recombination, which offered a greater opportunity for the more electron−hole pairs to participate in the redox reaction with H₂O. This result also agreed with the •O₂⁻ and •OH generation capability

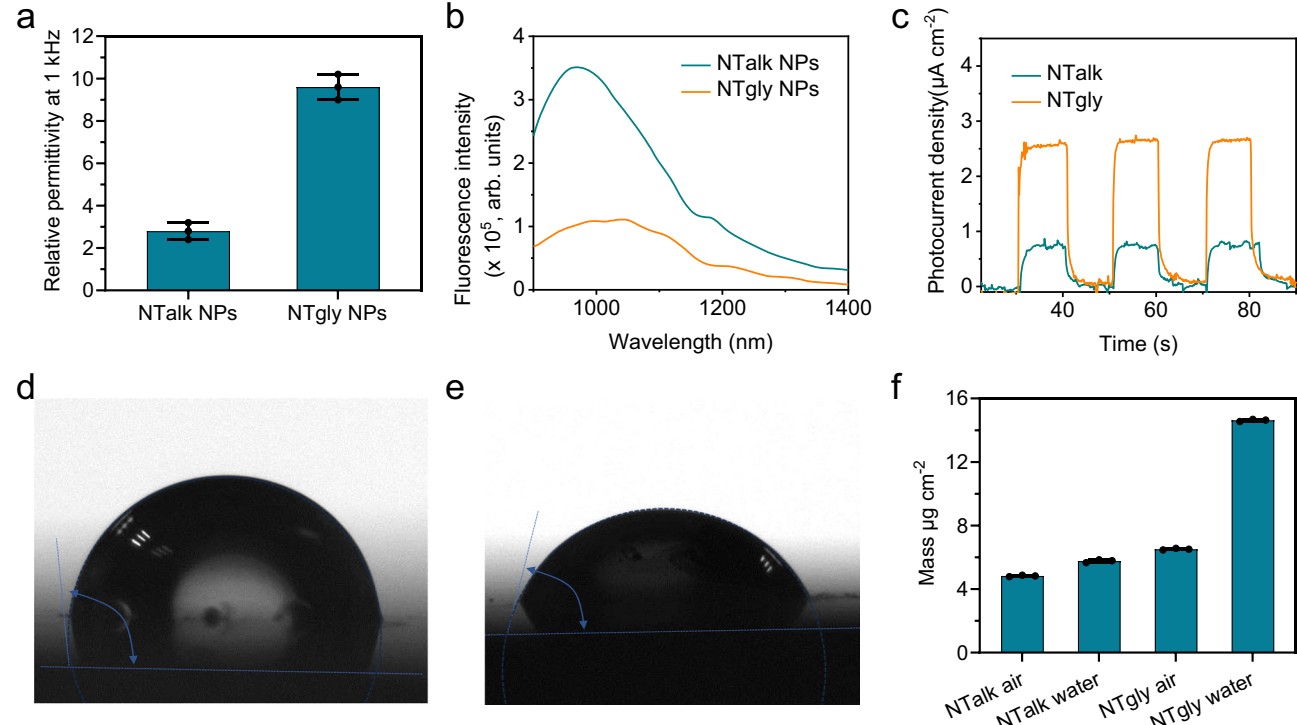

**Fig. 3 | The mechanism of precise side-chain engineering for optimizing 808 nm laser-driven generation of •$O_2^-$ and •OH. a** Relative permittivity at 1 kHz of NTalk and NTgly polymers. **b** Fluorescence spectra of NTalk NPs and NTgly NPs, excited at 808 nm (*n* = 3 independent samples; mean ± SD). **c** Transient photocurrent responses of NTalk and NTgly polymer film. Water contact angle measurements were performed on thin films of **d** NTalk and **e** NTgly polymers. **f** Quartz crystal microbalance measurements of the mass of thin films of NTalk and NTgly polymers suspended in air and water (*n* = 3 independent samples; mean ± SD).

(Fig. 2b, c). Next, we measured the water contact angles of NTalk and NTgly. As shown in Fig. 3d, e, the water contact angle of NTgly (70.7°) is lower than that of NTalk (108.2°), which indicates that glycol chains enhanced the wettability of NTgly polymers considerably in water, favoring $H_2O$ molecules to enter the PS NPs. Furthermore, quartz crystal microbalance measurements on thin films of NTalk and NTgly revealed that the mass of NTgly upon immersion in water was increased by 120% (Fig. 3f), which is sixfold of that for NTalk films, demonstrating that hydrophilic glycol oligomers as side chains could enable more $H_2O$ molecules to be brought to the proximity of hydrophobic PS skeletons and interact with the generated electron–hole pairs. These effects are consistent with more effective •$O_2^-$ and •OH generation for NTgly NPs (Fig. 2c) than that of NTalk NPs (Fig. 2b). These studies revealed that glycol side chains could optimize dynamic factors to boost •$O_2^-$ and •OH generation, contributing to ultra-low-power photoexcitation.

A possible mechanism is therefore proposed, which involves three processes: (1) PSs generate electron–hole pairs under 808 nm laser irradiation; (2) Photogenerated holes (high oxidation potential vs NHE > 0.81 V) oxidize $H_2O$ into $O_2$ and $H^+$; (3) A part of $O_2$ was reduced into •$O_2^-$ by photogenerated electrons. Meanwhile, another part of $O_2$ and $H^+$ were also rapidly reduced into •OH by photogenerated electrons. The PS skeleton design satisfies the thermodynamic and low bandgap conditions to realize NIR excitation and $O_2$-independent •$O_2^-$ and •OH production. The side-chain engineering of PS optimizes dynamic factors to improve •$O_2^-$ and •OH production efficiency to obtain ultra-low-power-density photoexcitation. Compared to the low collision probability of PSs with dissolved $O_2$ in water for the conventional type-I mechanism, hole-produced $O_2$ can be immediately reduced by photogenerated electrons, which boosts the production of •$O_2^-$ and •OH. In addition, the $H^+$ produced in the process also facilitates $O_2$ conversion into the most toxic •OH ($O_2 + 2H^+ + 2e^- \rightarrow 2$•OH), which could help improve the therapeutic outcomes. In the conventional type-I mechanism, the oxygen element of •$O_2^-$/•OH was derived from dissolved $O_2$ in water, and the hydrogen of •OH may derive from $H_2O$ or other molecules[37,69]. Therefore, the conventional type-I PSs still require tumor-endogenous $O_2$. To the best of our knowledge, this is the clear and detailed demonstration of $O_2$-independent pure organic PSs under ultra-low-power-density 808 nm excitation to generate •$O_2^-$ and •OH.

## Tumor-specific PSs

The studies of our proposed $O_2$-independent PSs are still in their infancy, and the currently reported PSs lack tumor specificity[39,40,54,70]. Therefore, activatable PSs that can be activated by tumor-related biomarkers (i.e., abnormal redox level, enzymes, pH) are very appealing for the specific killing of tumor cells. Recent reports have indicated that most cancer cells exhibit increased hydrogen peroxide ($H_2O_2$) levels that could be used to develop $H_2O_2$-activatable PDT strategies[71]. Based on the above analysis of mechanism, we herein developed an $H_2O_2$-activatable and tumor-endogenous $O_2$-independent PDT NP (BOH NP) for controllable •$O_2^-$ and •OH release under 808 nm laser irradiation at 15 mW cm$^{-1}$ (Fig. 4a). The BOH polymer was synthesized in two steps (Supplementary Fig. 33). We first used the naphthalene-diimide monomers with side chains of *N,N*-dimethylpropan-1-amine (35%) and glycol oligomers (65%) to polymerize with the bithiophene monomer (100%) (Supplementary Fig. 34). The $M_n$ and PDI of the polymer were 6.8 kDa and 2.05, respectively. Then (4-(bromomethyl) phenyl)boronic acid was further used to react with *N,N*-dimethylpropan-1-amine side chain of the above polymers to form amphiphilic BOH polymers. BOH polymers were self-assembled into BOH NPs in water with a spherical morphology and a size of 96 nm (Supplementary Fig. 35). BOH NPs can be converted to NMe NPs in response to $H_2O_2$ (Fig. 4a). Next, the optical properties of BOH NPs toward $H_2O_2$ were investigated. In the absence of $H_2O_2$, BOH NPs showed an absorption peak at 650 nm with negligible absorbance at 808 nm (Fig. 4b).

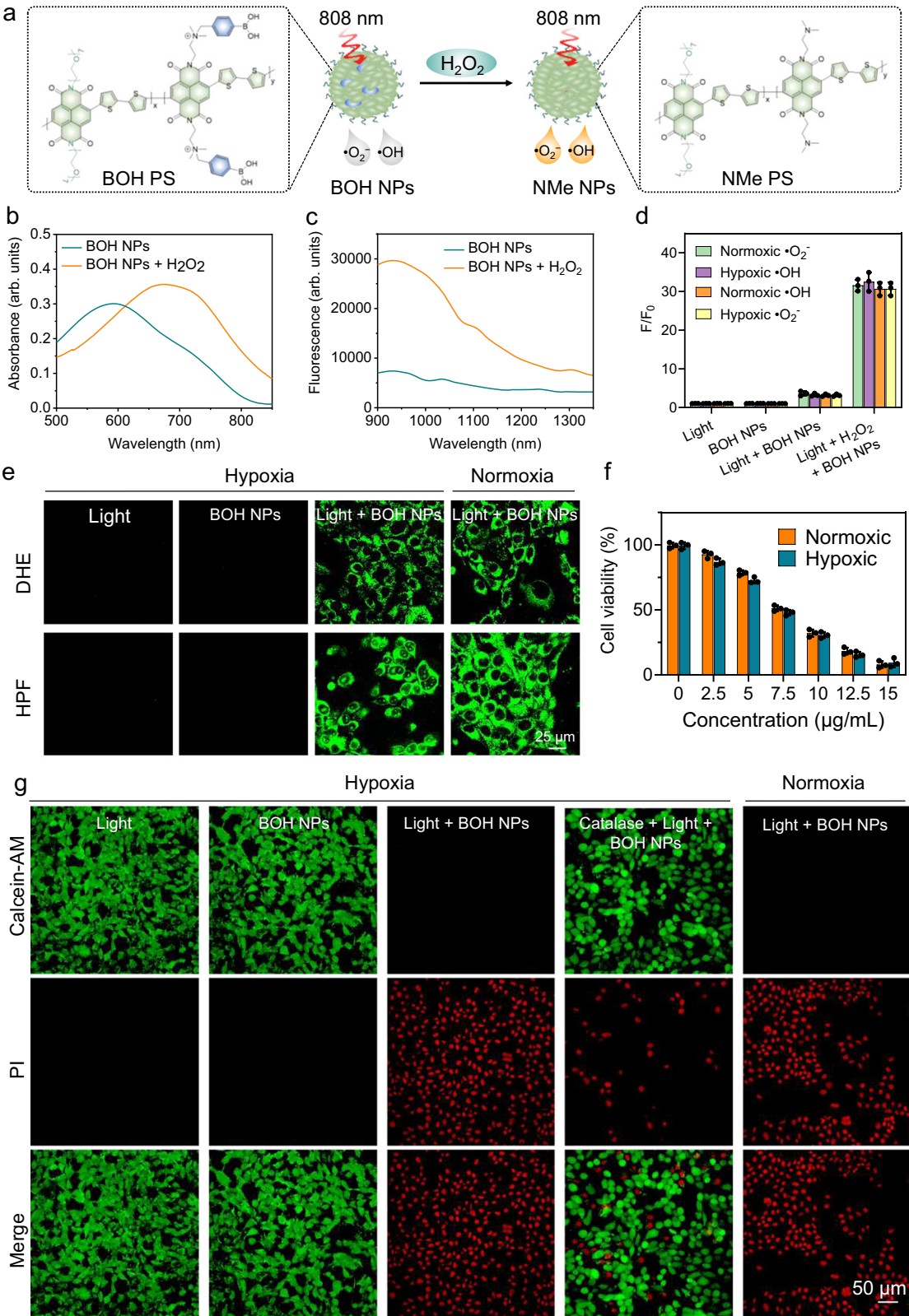

**Fig. 4 | Tumor-specific PS and in vitro cell experiment. a** The design and structure of $H_2O_2$-activatable PSs. The **b** absorption and **c** fluorescence spectra of BOH NPs (5 μg mL$^{-1}$) before and after response to $H_2O_2$ (10 μM, 35 min). **d** The fluorescence intensity changes of •$O_2^-$ probe (DHR 123, 2.0 μM) at 525 nm and •OH probe (HPF, 10 μM) at 515 nm for $H_2O_2$-activatable BOH NPs (5 μg mL$^{-1}$) ($n$ = 3 independent samples; mean ± SD). **e** ROS detection in 4T1 cells under normoxia and hypoxia conditions using DHE and HPF as •$O_2^-$ and •OH fluorescence indicators, respectively. **f** In vitro viability of 4T1 cells incubated with at different concentrations of BOH NPs solutions for 24 h under 808 nm irradiation at 15 mW cm$^{-2}$ for 5 min ($n$ = 3 biologically independent samples; mean ± SD). **g** Fluorescence images of calcein-AM (green, live cells) and PI (red, dead cells) co-stained 4T1 cells under different conditions. Each experiment was repeated three times with similar results.

By contrast, upon addition of $H_2O_2$, the absorption peak was red-shifted to 730 nm with strong absorbance at 808 nm. Therefore, the NIR-II emission peak at 940 nm was increased by approximately five-fold under the 808 nm laser excitation (Fig. 4c), which could be used for activatable NIR-II fluorescence imaging. The kinetics of $H_2O_2$-induced NIR-II fluorescence change of the BOH NPs (5 μg mL$^{-1}$) showed a nearly complete probe activation within 35 min in the presence of $H_2O_2$ (10 μM in pH = 7.4 PBS buffer) (Supplementary Fig. 36). The selectivity study indicated that only $H_2O_2$ could dramatically cause the 940 nm emission to increase for the BOH NPs (Supplementary Fig. 37).

Next, the $\cdot O_2^-$ and $\cdot OH$ generation capability of BOH NPs before and after the addition of $H_2O_2$ was evaluated using both $\cdot O_2^-$ fluorescent probe (DHR 123) and $\cdot OH$ fluorescent probe (HPF) in normal and hypoxic conditions. As shown in Fig. 4d, the $\cdot O_2^-$ and $\cdot OH$ generation capability of BOH NPs was increased over 10-fold after adding $H_2O_2$, respectively, which could benefit tumor-specific PDT. Further, Dihydroethidium (DHE) and HPF were used to evaluate the intracellular $\cdot O_2^-$ and $\cdot OH$ generation of BOH NPs in 4T1 cells under both hypoxic and normoxic conditions. 4T1 cells were incubated with BOH NPs and DHE or HPF probes, which were subsequently washed by PBS buffer and irradiated by an 808 nm laser (15 mW cm$^{-2}$, 5 min) before confocal imaging (Fig. 4e). Compared with light- or BOH NPs-treated cells, DHE and HPF probes in light + BOH NPs showed bright green fluorescence, indicating that $\cdot O_2^-$ and $\cdot OH$ were generated under both hypoxic and normoxic conditions in 4T1 cells. The cytotoxicity of the BOH NPs was further investigated in 4T1 cells by MTT. BOH NPs exhibited negligible dark cytotoxicity (Supplementary Fig. 38), at concentrations up to 200 μg mL$^{-1}$, indicating their good biocompatibility at the cellular level. BOH NPs showed dose-dependent phototoxicity under 808 nm irradiation (Fig. 4f). Over 95% of cells were killed in hypoxic conditions even at BOH NP concentrations as low as 10 μg mL$^{-1}$, demonstrating its great promise against tumor hypoxia. Additionally, the cytotoxicity of BOH NPs was determined using calcein-AM/propidium iodide (CA/PI) to distinguish the live and dead 4T1 cells. As shown in Fig. 4f, brighter red fluorescence of PI was observed from 4T1 cells treated with light and BOH NPs as compared to the cells treated with light or BOH NPs only (Fig. 4g), suggesting the stronger killing effect of BOH NPs on tumor cells. In addition, after catalase, a scavenger of $H_2O_2$, was used to clear the $H_2O_2$ in cells, only weak fluorescence of PI was observed from 4T1 cells treated with light and BOH NPs. These results demonstrated that BOH NPs could offer tumor $H_2O_2$-activatable $\cdot O_2^-$ and $\cdot OH$ generation, indicating their potential applications of tumor-specific PDT.

## NIR-II imaging of BOH NPs in vivo

Integrating high-resolution NIR-II fluorescence imaging and tumor-endogenous $O_2$-independent $\cdot O_2^-$ and $\cdot OH$ generation in a single probe holds great promise for noninvasive and precise NIR-II fluorescence imaging-guided PDT. To study the tumor-specific imaging capability of BOH NPs, BOH NPs were injected intravenously into 4T1 tumor-bearing mice and then time-dependent whole-body NIR-II fluorescence images were collected. The fluorescence of BOH NPs exhibited a gradual increase in tumor (Fig. 5a). The NIR-II fluorescent signal reached its maximum at 12 h and remained to be high until 36 h, indicating their successful tumor accumulation and effective $H_2O_2$ activation. To further verify that the increased NIR-II fluorescence originated from the action of $H_2O_2$ in tumor, N-acetyl-l-cysteine (NAC), an $H_2O_2$ scavenger, was injected into the tumor before the intravenous injection of BOH NPs. The fluorescence was gradually increased from 0 to 36 h. However, BOH NPs and NAC-treated tumors showed weaker brightness than that of only BOH NPs-treated group at the same time point (Fig. 5a, b), indicating that the $H_2O_2$ level plays an important role in activating BOH NPs. The ex vivo imaging of the excised organs and tumors further showed that BOH NPs exhibited bright fluorescence in tumor (Supplementary Fig. 39). Conclusively, these results confirmed that BOH NPs were suitable to be used as activatable probes for NIR-II fluorescence imaging-guided PDT.

## In vivo PDT evaluation of BOH NPs

Since BOH NPs showed superior oxygen-independent PDT effects in vitro, the tumor model we established was confirmed at a severe hypoxic state by immunofluorescence staining based on a hypoxia-associated protein, hypoxia-inducible factor 1-α (HIF-1α) (Fig. 5c). Besides, tumor blood vessels were further labeled using an anti-CD31 antibody (Fig. 5c). As is well-known, oxygen is mainly transported by blood vessels in vivo. However, only a limited number of tumor blood vessels (red) were found. By contrast, the normal tissues from adjacent tumor margins exhibited more prominent vasculature (depicted in red) and less presence of HIF-1α (depicted in green) compared to the tumor tissues (Supplementary Fig. 40), which demonstrated tumor hypoxia. Next, the phototoxicity of BOH NPs for tumor in vivo was further evaluated in 4T1 tumor-bearing mice. 4T1 tumor-bearing female Balb/c mice with a tumor volume of 60 mm$^3$ were randomly divided into three groups ($n = 5$), including (i) PBS group, (ii) BOH NPs group, the 4T1 tumor-bearing mice were only intravenously injected with BOH NPs (5 mg kg$^{-1}$) without irradiation; (iii) 808 nm laser group, the mice were only irradiated under 808 nm laser for 10 min; (iv) BOH NPs + 808 nm laser group, the 4T1 tumor-bearing mice was intravenously injected with BOH NPs (5 mg kg$^{-1}$), then with 808-nm laser irradiation at 12 h postinjection. The tumor temperatures of the BOH NPs + 808 nm laser group were real-time monitored by the IR thermography, which only showed an enhancement of less than 1 °C after 10 min irradiation (Supplementary Fig. 41), indicating that the therapeutic effect was entirely due to PDT rather than photothermal effect. The body weight of the mice in each group showed no significant change during the 14-day treatment period (Fig. 5d), suggesting that PDT process exhibited negligible adverse effects on the health of mice. The change in tumor volume was monitored to evaluate the therapeutic effects. As shown in Fig. 5e, the tumor volume of 808 nm laser group and BOH NPs group increased remarkably. In contrast, the tumor in BOH NPs + 808 nm laser group was eliminated on the 8th day of treatment, and even after 12 days, no obvious tumor recurrence was found (Fig. 5e). The representative tumor images (Fig. 5f) and average tumor weight (Fig. 5g) further demonstrated the excellent antitumor efficacy of BOH NPs with light irradiation. The Hematoxylin and eosin (H&E) staining and terminal deoxynucleotidyl transferase dUTP nick end labeling (TUNEL) immunofluorescence staining assay further verified remarkable antitumor efficacy of BOH NPs with irradiation of 808 nm laser (Fig. 5h). Moreover, undamaged main organs were detected by H&E staining, which demonstrated good biosafety of the treatment (Supplementary Fig. 42).

To fully demonstrate the advantages of BOH NPs in overcoming tumor hypoxia and tissue penetration, we established a mice model bearing a larger tumor (~500 mm$^2$) (Supplementary Fig. 43a). Immunofluorescence imaging of tumor HIF-1α (Supplementary Fig. 43b) indicates that the tumor of ~500 mm$^2$ shows more obvious hypoxia than the tumor of ~60 mm$^2$ because tumor hypoxia is positively related to the size of the tumor[72]. The mice underwent the same treatment procedures as those with tumors of ~60 mm$^2$. During treatment, the body weight of mice in the treatment group was stable (Supplementary Fig. 43c). As shown in Supplementary Fig. 43d, the tumor volume of 808 nm laser group and BOH NPs group increased remarkably until about 1600 mm$^3$. In contrast, the tumor volume in BOH NPs + 808 nm laser group gradually decreased and finally maintained at ~150 mm$^3$. The corresponding tumor photographs (Supplementary Fig. 43e) and average tumor weights (Supplementary Fig. 43f) exhibit prominent tumor suppression with a tumor growth inhibition rate of 91%. The H&E and TUNEL immunofluorescence assays reaffirmed the significant antitumor efficacy of BOH NPs under 808-nm laser irradiation (Supplementary Fig. 43g). These findings underscore the pronounced

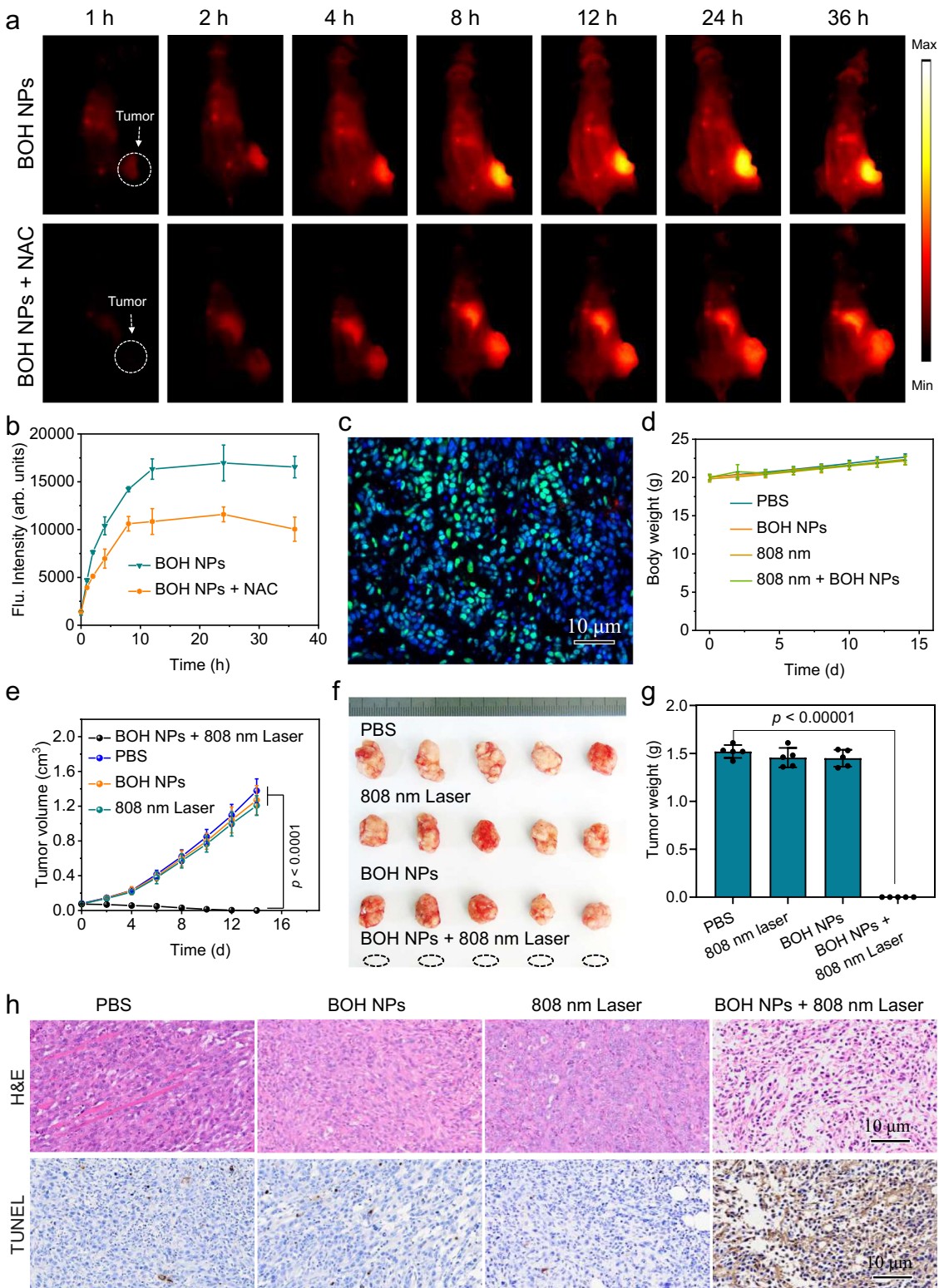

O$_2$-independent PDT effect and deep-tissue penetration at 808 nm of BOH NPs, particularly effective against deep hypoxic tumors.

We further evaluated the side effects of treatment. At the end of the treatment, we conducted thorough investigations of the physiological parameters in mice. Hematological data analysis (Supplementary Fig. 44), blood biochemical assays (Supplementary Fig. 45), and histological examinations of vital organs (Supplementary Fig. 46) collectively indicate that the treatment does not induce obvious infection and inflammation, adverse effects on basic liver and kidney,

and apparent histological abnormalities or lesions in the treated mice, indicating no obvious side effects for the treatment of BOH NPs.

As the biosafety testing of PSs is critical for in vivo treatment, we further investigated the in vivo toxicology of PSs. Standard analyses, including hematological, blood biochemical, and histological assessments at time points 1, 7, and 30 days postinjection of BOH NPs (injection dose: 100 mg Kg$^{-1}$), were executed. Mice administered BOH NPs exhibited no discernible decrease in body weight compared to control mice treated with saline (Supplementary Fig. 47a).

**Fig. 5 | In vivo NIR-II fluorescence imaging and PDT evaluation of BOH NPs.**
**a** Representative time-dependent in vivo NIR-II fluorescence images of 4T1 tumor-bearing mice after intravenous injection of PBS (i.p., 25 μL) or NAC (i.p., 10 mg/kg in 25 μL of PBS) followed by intravenous injection of BOH NPs (1 mg mL$^{-1}$, 100 μL) at $t = 6$ h later, $n = 3$ mice per group. **b** Quantifying NIR-II fluorescence intensities of tumor tissue of 4T1 tumor-bearing mice after intravenous injection of BOH NPs in a different time ($n = 3$ biologically independent samples; mean ± SD).
**c** Immunofluorescence imaging of tumor slices. The tumor blood vessels (red) are stained with the anti-CD31 antibody, hypoxia-related protein HIF-α is stained with the anti-HIF-1α antibody (green). The (**d**) body weights and **e** tumor growth curves

of the mice in vivo PDT study after intravenous injection of BOH NPs (1 mg mL$^{-1}$, 200 μL). The tumors of BOH NPs and BOH NPs + 808 nm groups were irradiated by an 808 nm laser with a power density of 15 mW cm$^{-2}$ for 10 min. Mean ± SD, $n = 5$. $P < 0.0001$. Statistical significance was determined using one-way ANOVA. **f** Photos of tumors collected from the mice in different groups at the end of PDT. **g** The mean weight of tumors separated from mice after different treatments. $p < 0.0001$ ($n = 5$ independent samples; mean ± SD). Statistical significance was determined using one-way ANOVA. **h** Representative H&E- and TUNEL-stained slices of tumors were collected from 4T1 tumor-bearing mice on the sixth day of the treatments in vivo PDT study.

Additionally, the mice did not show any significant differences in behaviors and appearance, including eating, drinking, hair color and glossiness, between the BOH NPs-treated mice and the control groups (Supplementary Fig. 47b). Hematological indexes in the BOH NPs-treated groups showed minimal differences compared to the saline group at all time points (Supplementary Fig. 48), suggesting an absence of apparent infection and inflammation in the treated mice. Furthermore, there were no significant differences in blood bio-chemical parameters observed between the groups treated with BOH NPs and the saline group at all examined time points (Supplementary Fig. 49), indicating no presence of hepatotoxicity, nephrotoxicity, or hematological toxicity. Lastly, immunohistochemistry revealed no apparent histological abnormalities or lesions in major organs (Supplementary Fig. 50). These findings collectively affirm that BOH NPs demonstrate no evident toxicity in vivo.

## Discussion

We synthesized organic semiconducting polymers as PSs for $O_2$-independent PDT applications upon 808 nm irradiation with ultra-low-power density. Such mechanism was revealed to include three steps: (i) PSs generate electron–hole pairs under 808 nm laser irradiation; (ii) Photogenerated holes oxidize $H_2O$ into $O_2$ and $H^+$. (iii) $O_2$ and $H^+$ were rapidly reduced into •OH by photogenerated electrons. Meanwhile, some $O_2$ was reduced into •$O_2^-$ by photogenerated electrons. The PS skeleton design satisfies the thermodynamic and low bandgap conditions to realize NIR excitation and $O_2$-independent •$O_2^-$ and •OH production. The side-chain engineering of PS optimizes dynamic factors to improve •$O_2^-$ and •OH production efficiency, which made the ultra-low-power-density photoexcitation possible.

Unlike Type-I PSs, our PSs do not require tumor-endogenous $O_2$ to participate in the reaction for •$O_2^-$ and •OH generation. Rather, the semiconducting polymers can directly sensitize $H_2O$ into •$O_2^-$ and •OH under 808 nm laser irradiation, enabling completely tumor-endogenous $O_2$-independent PDT. Existing $O_2$-independent PSs need to transfer electrons to oxidizable substrates (e.g., disulfide bonds of proteins) and holes into intracellular reducible substrates (e.g., GSH and NADH). Compared with these locally distributed and exhaustible redox substrates in tissues, $H_2O$ is most abundant and ubiquitous, which is ideal for the efficient production of •$O_2^-$ and •OH for PDT. In addition, the proximity between the PS and reaction substrates is a crucial dynamic factor determining the reaction rate. The limited propagation distance of electron–hole pairs in water imposes a constraint on the spatial range for reactions. In contrast to the small size and large amount of water molecules within the tissues we utilized, biological molecules, such as GSH, NADH, nucleobases and amino acid residues, possess larger size and volume. Consequently, their penetration and reactivity within the nanoscale particles of the PS present notable challenges, which reduce the photodamage for normal tissues, limiting the side effects to a certain extent. Finally, as compared to the random design of reported $O_2$-independent PSs, we present a smart and versatile strategy to design our PS skeleton through the rational coupling of electron donors and acceptors, both with low HOMO levels.

To mitigate electron–hole recombination for PSs, previous reports have focused more on designing PS skeletons to facilitate the inter-system crossing process or using locally distributed and exhaustible substances, such as GSH and NADH, to scavenge holes. In addition, the PDT-related photosensitized reaction of the currently reported PSs mainly occur on the surface of the PS nanoparticles with substances, such as GSH and NADH, due to the strong hydrophobicity of PS-conjugated skeletons and alkyl side chains, and the large size of substances[45,58,63]. By contrast, we provided a simple strategy of attaching glycol oligomers instead of alkyl groups as side chains to organic PS skeletons. Detailed characterization verified that glycol side chains could boost the permittivity of PSs to inhibit electron–hole recombination. As compared to the hydrophobic alkyl chains, hydrophilic glycol side chains could also bring more $H_2O$ molecules into the PS vicinity to interact with electron–hole pairs generated from the PS skeletons in PS nanoparticles. Together, these effects optimized the dynamic factors to improve •$O_2^-$ and •OH generation, and the optimized NTgly NPs offered 51-fold stronger •$O_2^-$ and •OH production than that of clinically approved MB. This yielded an ultra-low-power-density 808 nm photoexcitation at 15 mW cm$^{-2}$ for PDT of tumor treatment.

PDT has shown great promise for the treatment of many cancers. However, its clinical progress is impaired by the safety concern associated with the target specificity of PSs. Due to the limitation of design and synthesis for our mechanism PSs, the studies of such PSs are still in their infancy, not to mention target-specific PSs based our mechanism. Therefore, we developed an activatable PS that specifically responds to the high $H_2O_2$ level in tumors, which improves the PDT specificity and reduces the side effects. Our study thus provides a design guideline to develop PSs toward $O_2$-independent •$O_2^-$ and •OH generation under deep-tissue-penetrating NIR photoexcitation with ultra-low-power density and short irradiation time by pure organic materials. More importantly, our proposed organic PSs have great promise to advance PDT platforms into clinical trials.

## Methods

The research presented here complies with all relevant ethical regulations. All experiments involving animals were reviewed and approved (IACUC-002-39) by the guidelines of the Laboratory Animal Center of Jiangsu KeyGEN BioTECH Corp., Ltd., prior to commencing the study. All the maximal tumor size/burden in our experiments did not exceed the maximal tumor size/burden permitted (2000 mm$^3$) by the Laboratory Animal Center of Jiangsu KeyGEN BioTECH Corp., Ltd.

### Preparation of NPs

NPs were prepared by using a nanoprecipitation method. In a typical preparation, the conjugated polymers, NTalk, NTgly and NTOalk, and the amphiphilic polymer PSMA were dissolved in anhydrous THF to prepare a stock solution containing conjugated polymers (0.1 mg mL$^{-1}$), and the amphiphilic polymer PSMA (0.02 mg mL$^{-1}$). A 5-mL aliquot of this solution was quickly dispersed into 10 mL of Milli-Q water under vigorous sonication. The solution was blown nitrogen gas to remove THF at 37 °C for 48 h. Before use, any precipitate in the aqueous solution was removed by a 0.2-μm cellulose membrane filter. The aqueous solution underwent triple washing with distilled–deionized water.

After each wash, centrifugation was carried out using 50 K centrifugal filter units from Millipore at a speed of 136.975×$g$ for 10 min at a temperature of 10 °C.

## ROS test

•$O_2^-$ test: DHR 123 was used as the •$O_2^-$ fluorescence probe. DHR 123 (10.0 μM) and NPs (5 μg mL$^{-1}$) were prepared in the aqueous solution in normoxia (21% $O_2$) or hypoxic conditions (hypoxic conditions established in a glovebox with <0.01 ppm $O_2$). After the 808 nm laser irradiated with the power density of 15 mW cm$^{-2}$ for 5 min, the fluorescence spectra of the solution were measured. The fluorescence spectra of DHR 123 with only 808 nm laser irradiation or only NP treatment were used as the control ($Ex$ = 480 nm). The molal concentration of MB and NTgly is equal. [MB] = 0.1 μg/mL.

•OH test: HPF was used as the •OH fluorescence probe. The mixture of HPF (10.0 μM) and NPs (5 μg mL$^{-1}$) was prepared in the aqueous solution in normoxia (21% $O_2$) or hypoxic (1% $O_2$) conditions. After the 808 nm laser irradiated with the power density of 15 mW cm$^{-2}$ for 5 min, the fluorescence spectra of the solution were measured. The fluorescence spectra of HPF with only 808-nm laser irradiation or only NP treatment were used as the control ($Ex$ = 480 nm). The molal concentration of MB and NTgly is equal. [MB] = 0.1 μg/mL.

## Intracellular •$O_2^-$ and •OH detection

•$O_2^-$ detection: 4T1 cells at a density of $5 \times 10^4$ cells per well were incubated with BOH NPs (5 μg mL$^{-1}$) for 12 h, followed by incubation with 10 μM DHE for another 30 min. Cells were washed with PBS buffer three times. Then, cells were irradiated with an 808 nm laser at a power density of 15 mW cm$^{-2}$ for 5 min in hypoxia or normoxia. Then the cells were immediately observed using CLSM with the excitation wavelength of 480 nm, and emission collection wavelength from 500 to 650 nm.

For •OH detection, 4T1 cells at a density of $5 \times 10^4$ cells per well were incubated with NPs (5 μg mL$^{-1}$) for 12 h followed by incubation with 10 μM HPF for another 30 min. Cells were washed with PBS buffer three times. Then the cells were irradiated with an 808 nm laser at a power density of 15 mW cm$^{-2}$ for 5 min in hypoxia or normoxia. The green fluorescence was immediately observed using CLSM with an excitation wavelength of 480 nm, and emissions were collected from 500 to 650 nm.

## In vivo therapeutic studies

All mice used in this study were maintained in a dedicated pathogen-free animal facility at 60% of humidity and 25 °C with 12/12 light schedule and free access to food and water. Female Balb/c mice bearing 4T1 tumors with a tumor volume of 60 mm$^3$ were randomly divided into 4 groups ($n$ = 8), including (i) PBS (ii) BOH NPs group, the 4T1 tumor-bearing mice were only intravenously injected with BOH NPs (1 mg mL$^{-1}$, 200 μL) without irradiation of 808 nm laser; (iii) 808 nm laser group, the mice was only irradiated under 808 nm laser with ultra-low-power density (15 mW cm$^{-2}$) for 10 min at postinjection 12 h; (iv) BOH NPs + 808 nm laser group, the 4T1 tumor-bearing mice were intravenously injected with BOH NPs (1 mg mL$^{-1}$, 200 μL), then 808 nm laser irradiation with ultra-low-power density (15 mW cm$^{-2}$) and short time (10 min) was performed at postinjection 12 h. During 14 days, mice were observed for changes in tumor growth and body weight. Tumor advancement was tracked by measuring individual tumor sizes using a digital vernier caliper. The formula used to calculate the individual tumor volume (**V**) was as follows: $V = (a \times b^2)/2$. In this formula, '$a$' represented the longer dimension of the tumor tissue, while '$b$' represented the shorter dimension perpendicular to the length. Three animals from each group were randomly selected for H&E- and TUNEL-stained slices of tumors on the sixth day of the treatments in the in vivo PDT study.

## Reporting summary

Further information on research design is available in the Nature Portfolio Reporting Summary linked to this article.

## Data availability

All the data supporting the findings of this study are available within the article and its Supplementary information files. The full image dataset is available from the corresponding author upon request. Source data are provided with this paper.

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

## Acknowledgements
This study is supported by the Singapore National Research Foundation (L.B., A-0009163-01-00, A-8000223-01-00, A-0009201-01-00).

## Author contributions
Y.T., Q.F. and B.L. conceived the concept of the study and designed the experiments. Y.T. and Y.L. performed the experiments, and W.B.L. and W.S. helped in material characterizations. Y.T., Y.L., W.B.L. and W.S. performed the in vitro and in vivo experiments. J.T., G.Q. and W.H. helped and gave valuable suggestions. Y.T., Q.F. and B.L. analyzed the experimental data and co-wrote the manuscript. All the authors discussed, commented, and agreed on the paper.

## Competing interests
The authors declare no competing interests.
