## [Peer Review File · Nature Communications]

Reviewers' Comments:

Reviewer #1:

Remarks to the Author:

The manuscript submitted by Tang et al. presents oxygen-independent type-III photosensitizers for tumor PDT under ultralow-power NIR photoexcitation. With the rational photosensitizer skeleton design and precise side-chain engineering, the generations of $O_2^{\bullet-}$ and $\bullet OH$ under oxygen-free conditions are realized using 808 nm photoexcitation. Although the work demonstrated here is novel and meaningful, both the innovation and scientific rigor of the manuscript are below the standard set by Nature Communications, thus I recommend this manuscript be published elsewhere.

1. In the authors' opinion, the mechanism of the photosensitizer works with an oxygen-independent pattern can be classified as the Type-III photosensitive mechanism. However, water is not suitable to be defined as targeting biomolecules. It seems more appropriate that the working mechanism of the current photosensitizers is an oxygen-independent Type-I electron transfer mechanism, resulting in the generations of $O_2^{\bullet-}$ and $\bullet OH$ with the sources of water and oxygen.
2. In the main text, the $O_2^{\bullet-}$ detections are totally based on DHR123 probe. However, there is no direct evidence to demonstrate that DHR123 is the $O_2^{\bullet-}$ specific probe. Instead, Dihydroethidium (DHE) may be a better choice.
3. I am confused that all the units of oxidation and reduction potentials are eV in this manuscript, should not that be V?
4. It is quite surprising that $\bullet OH$ can be generated through the reduction of O_2 under a such weak acid environment (< -0.473 eV vs. NHE, $O_2 + H^+ + e^- \rightarrow \bullet OH$, pH = 6.8), please offer more references to support this viewpoint.
5. Please give the references to support the viewpoint on Page 18 lines 349-352.
6. Please offer the oxygen concentration and the method for constructing a hypoxic environment in biological tests.
7. To evaluate whether the designed photosensitizers are capable of destroying the tumors in an oxygen-independent manner, the tested tumors should have endogenous hypoxic environments. However, in this paper, the PDT experiments were carried out only when the volume of tumors was around 60 mm³, which largely decreased the credibility and the significance of the main purpose mentioned in the title and abstract.

Reviewer #2:

Remarks to the Author:

This manuscript (NCOMMS-23-37329) investigates new oxygen independent photosensitizers activated by ultralow-power near-infrared (NIR) excitation for photodynamic therapy. The manuscript is well organized and the experiments have been carefully designed and carried out. The idea and the aims of the work are interesting and the development of new photosensitizers deserves efforts and scientific investment. However, some relevant aspects are unclear or poorly discussed. Therefore, I recommend this manuscript for publication in Nature Communications, after some corrections. Some comments that, in my opinion, should be considered in a revised version:

- The classification of mechanisms is wrong or, at least, in conflict with some reviews recently published that should be taken into account to establish the types of mechanism. Please, see the following review articles: Photochem. Photobiol. 93, 912-919 (2017); Photochem. Photobiol. 97, 1456-1483 (2021); Photochem. Photobiol. 99, 313-334 (2023).
- Superoxide radical anion and hydroxyl radical are not efficient species for PDT. Again, revise the articles mentioned in the previous paragraph and references cited therein and hypothesize why the photosensitizers works well although they produce species with low reactivity (superoxide) or with very low lifetime (hydroxyl radical).
- It would seem that the photosensitizer might react directly with biological substrates, for example, oxidizable biomolecules like nucleobases or amino acid residues. These pathways might contribute to the photodamage. This point is not discussed in detail and I think it is worth considering it.

Reviewer #3:

Remarks to the Author:

The study by Tang et. al. demonstrated an oxygen-independent PDT agent based on developing novel semiconducting polymers by rational skeleton design and precise side-chain engineering. It was verified that 808 nm photoexcitation can sensitize H₂O into •O₂⁻ and •OH under oxygen-free conditions. The investigation of photooxidation mechanisms with ¹⁸O-labelled mass spectra is interesting and convincing. In the biological tests, this system showed satisfying anti-cancer effect under a low-power laser irradiation and good selectivity of tumor imaging by responding to H₂O₂. However, improvements can be seen if the author can address the following issues:

Major concerns:

1. It was well-known that photosensitizers should be activated to its singlet states and triplet states for sensitizing oxygen or other substrates. It should be necessary to characterize the excited states of PSs by both theoretical methods and experimental methods, for example, transient spectrum study, energy calculations, and lifetime measurements of the singlet/triplet states. Also, it is exciting that the reported material can work under low-power 808 nm laser, but the reasons remain unclear. The authors are suggested to provide absorption coefficient, quantum yield of ROS or any other helpful information to further showcase the advantages of this PS as compared to the existing NIR PSs.
2. In Fig. 1g and Fig. S19, it was demonstrated that the NTgly showed a much larger red shift after self-assembly. Compared to NTalk (red-shifted 50 nm) and NTOalk (red-shifted 40 nm), NTgly showed a much larger red shift (150 nm). Thus, the reason may not only be aggregation-induced red shift. The authors should provide more information to explain this phenomenon.
3. There is a lack of biosafety testing. The dark toxicity of the NPs should be studied. Besides, after in vivo treatments, some physiological parameters of mice should be investigated.

Minor points:

1. In Fig 5a, BOH NPs have less liver accumulation than BOH NPs + NAC, why?
2. In Fig. 5c, it is necessary to show the CD31 and HIF levels for the control groups.
3. Balb/c should be revised as BALB/c.
4. In Line 576, the author mentioned "the molal concentration of MB and NTgly is equal". How to determine the molal concentration of NTgly?

December 20, 2023

Dear Reviewers

We appreciate the reviewers' very helpful comments and revised the manuscript accordingly. A point-by-point reply was included below. All the changes are marked with red color in the revised manuscript for your reference. Thank you very much for your kind attention.

Our responses to the Reviewer's comments are summarized as follows:

COMMENTS TO AUTHOR:

Reviewer #1 (Remarks to the Author):

The manuscript submitted by Tang et al. presents oxygen-independent type-III photosensitizers for tumor PDT under ultralow-power NIR photoexcitation. With the rational photosensitizer skeleton design and precise side-chain engineering, the generations of $O_2^{\bullet-}$ and $\bullet OH$ under oxygen-free conditions are realized using 808 nm photoexcitation. Although the work demonstrated here is novel and meaningful, both the innovation and scientific rigor of the manuscript are below the standard set by Nature Communications, thus I recommend this manuscript be published elsewhere.

Reply: We are grateful for the Reviewer's valuable feedback on our manuscript, which will definitely help us further improve the quality of this work. We appreciate the Reviewer's positive comment that the work demonstrated here is novel and meaningful. We also take the Reviewer's concerns seriously. We have performed literature investigation and some experiments to support and highlight our study, and we sincerely hope that you will now accept the revised version of the paper for publication.

Conventional photodynamic therapy (PDT) based on existing Type-I or Type-II photosensitizers (PSs), has limitations due to oxygen-dependent photosensitization pathways and high-power-density photoexcitation. Herein, we report an organic PS with oxygen-independent ROS production and ultralow-power-density 808 nm photoexcitation. Deviating from both existing Type I and Type II photosensitizers, the new organic PS sensitizes H_2O to achieve O_2 -independent $\bullet O_2^-$ and $\bullet OH$ production. The combination of rational PS skeleton design and precise side-chain engineering synergistically contribute to the elevated $\bullet O_2^-$ and $\bullet OH$ production efficiency of our PSs, which is over 50-fold stronger than that of clinically approved methylene blue (MB). Activatable O_2 -independent PSs were further developed for *in vivo* tumor-specific photodynamic ablation under 808 nm irradiation with an ultralow-power of 15 mW cm^{-2} to yield highly efficient PDT. Detailed photophysical mechanism studies further provided insights into the new PDT mechanism. We believe that our findings will advance PS development.

1. In the authors' opinion, the mechanism of the photosensitizer works with an oxygen-independent pattern can be classified as the Type-III photosensitive mechanism. However, water is not suitable to be defined as targeting biomolecules. It seems more appropriate that the working mechanism of the current photosensitizers is an oxygen-independent Type-I electron transfer mechanism, resulting in the generations of $O_2^{\bullet-}$ and $\bullet OH$ with the sources of water and oxygen.

Reply: We thank the Reviewer for the valuable comments on the classification of photosensitizer

mechanisms. The Type-III photosensitive mechanism, as discussed in Laustriat's work ("Molecular mechanisms of photosensitization," *Biochimie*, 1986), encompasses photosensitized reactions that do not involve oxygen. The original of this paper is "...Photosensitized reactions which do not involve oxygen (Type III reactions). Such reactions need specific conditions, *i.e.* deaerated systems or high local concentrations of **reactants** - as occur in organized systems - **in order to bypass the oxygen effect**....". Notably, the term "reactants" used in this context encompasses various molecules, such as targeting biomolecules, water, or other molecular species, as we have previously discussed in our manuscript. In addition, another relevant literature (Hamblin MR, Abrahamse H. "Oxygen-Independent Antimicrobial Photoinactivation: Type III Photochemical Mechanism?" *Antibiotics (Basel)* 9, (2020)) also discussed Type III photosensitive mechanism. The paper says "...The purpose of the present review is to highlight the possibility of **oxygen-independent photoinactivation** leading to the killing of pathogenic bacteria, which may be termed the "**Type III photochemical pathway**", ... we therefore propose that oxygen-independent photoinactivation of microbial species as the "**Type III photochemical pathway**"....". This publication also did not specify **targeting biomolecules**. Based on these references, we defined our mechanism as the Type-III photosensitive mechanism in our previous manuscript.

However, we've also noticed that the literature, specifically a paper (Yao Q, et al. The concept and examples of type-III photosensitizers for cancer photodynamic therapy. *Chem* 8, 197-209 (2022)), emphasized the inclusion of targeting biomolecular in the definition of Type-III photosensitizers. The original statement of the paper is quoted here "there might be a third photosensitive mechanism acting without ROS intermediates (type-III mechanism). In this **type-III mechanism, the photosensitizer itself should have a specific targeting property to proteins, nucleic acids, and other bio-macromolecules** in cells".

After more literature review, it becomes obvious that the research community does not yet have a unified definition of Type III photosensitizers. To avoid confusion, we have decided to refer to our mechanism as a new mechanism without assigning a specific name at this time. Our PSs generate $\bullet\text{O}_2^-$ or $\bullet\text{OH}$ through electron and hole transfer between the excited PSs and adjacent H_2O , **eliminating the reliance on O_2** , which is different from both Type I and Type II.

Therefore, we deleted "type-III" in the revised manuscript.

2. In the main text, the $\text{O}_2\bullet^-$ detections are totally based on DHR123 probe. However, there is no direct evidence to demonstrate that DHR123 is the $\text{O}_2\bullet^-$ specific probe. Instead, Dihydroethidium (DHE) may be a better choice.

Reply: Thanks for your professional suggestion. In response to the Reviewer's insightful suggestion, we have conducted additional cell experiments utilizing Dihydroethidium (DHE). The results, presented in Figure R1, reveal that the fluorescence enhancement observed with DHE probes is nearly identical to that of DHR123. The additional cell experiment data utilizing DHE has replaced the DHR123 data of Fig 4e in the revised manuscript (Page 22, Line 415).

Figure R1. $\bullet\text{O}_2^-$ detection in 4T1 cells under normoxia and hypoxia conditions using DHE as $\bullet\text{O}_2^-$ fluorescence indicators.

3. I am confused that all the units of oxidation and reduction potentials are eV in this manuscript, should not that be V?

Reply: Thank you for your valuable advice. The unit should be V not eV. We have corrected the mistake in the revised manuscript.

4. It is quite surprising that $\bullet\text{OH}$ can be generated through the reduction of O_2 under a such weak acid environment (< -0.473 eV vs. NHE, $\text{O}_2 + \text{H}^+ + \text{e}^- \rightarrow \bullet\text{OH}$, pH = 6.8), please offer more references to support this viewpoint.

Reply: We appreciate the Reviewer's comment. We derived the potential corresponding to a pH of 6.8 based on the available literature (Angew. Chem. Int. Ed. 2019, 58, 632–636), which does not seem to be right. During the revision, we read more literature and now revised the value of pH=6.8 to the more widely documented value of pH=7.0 (0.31 V vs. NHE, $\text{O}_2 + \text{H}^+ + \text{e}^- \rightarrow \bullet\text{OH}$, pH = 7) in the revised manuscript. (References: 1. Sawyer DT. The redox thermodynamics for dioxygen species (O_2 , O_2^- , HO_2^- , HO_2H , and HO_2^-) and monooxygen species (O , O^- , $\bullet\text{OH}$, and $-\text{OH}$) in water and aprotic solvents. *Basic Life Sci* 49, 11-20 (1988). 2. Wood PM. The potential diagram for oxygen at pH 7. *Biochem J* 253, 287-289 (1988). 3. Nordberg M, Duffus JH, Templeton DM. Explanatory dictionary of key terms in toxicology (IUPAC Recommendations 2007). *Pure Appl Chem* 79, 1583-1633 (2007). 4. Hoare, J. P. (1985) in *Standard Potentials in Aqueous Solution* (Bard, A. J., Parsons, R. & Jordan, J., eds.), pp. 49-66, IUPAC/Marcel Dekker, New York.)

The potential has been revised to the value of pH=7.0 in the main text (Page 3, Line 71; Page 6, Line 104; Page 7, Line 131 and Page 13, Line 266).

5. Please give the references to support the viewpoint on Page 18 lines 349-352.

Reply: Thank you very much for your advice. We have added more references, including 1. Laustriat G. "Molecular mechanisms of photosensitization." *Biochimie* 68, 771-778 (1986); 2. Sharman, Wesley M., Cynthia M. Allen, and Johan E. van Lier. "Role of activated oxygen species in photodynamic therapy." *Methods in enzymology* 319, 376-400(2000)., related to the viewpoint on Page 18 lines 349-352. The related references have been added to the main text as references 37 and 69 (Page 18, Line 348).

6. Please offer the oxygen concentration and the method for constructing a hypoxic environment in biological tests.

Reply: We acknowledge the valuable feedback from the Reviewer and sincerely apologize for our

oversight. To construct a hypoxic environment in biological tests, we employed an AnaeroPack™-Anaero Anaerobic Gas Generator within a sealed transparent chamber. Cells were precultured in this controlled environment for 12 hours at 37 °C before initiating ROS detection. The ROS-ID™ hypoxia/oxidative stress detection kit was utilized in accordance with the product manual to signify the intracellular hypoxic condition. During the experiment, the co-incubation of BOH-NPs and the ROS detection probe with cells occurred within the hypoxic setting, with an oxygen concentration of approximately 1%. All other procedures remained consistent with those performed in a normoxic environment. The description has been added to Supporting Information (Page S36, Lines 632-640).

7. To evaluate whether the designed photosensitizers are capable of destroying the tumors in an oxygen-independent manner, the tested tumors should have endogenous hypoxic environments. However, in this paper, the PDT experiments were carried out only when the volume of tumors was around 60 mm³, which largely decreased the credibility and the significance of the main purpose mentioned in the title and abstract.

Reply: We appreciate the Reviewer's comment about this point. To evaluate whether the designed photosensitizers are capable of destroying the tumors in an oxygen-independent manner, we first investigated photooxidation mechanisms with ¹⁸O-labelled mass spectra in vitro (Fig. 2h-2i in the main text), which demonstrated that the oxygen element of ROS was derived from H₂O rather than dissolved O₂ in water. The isotopic mass spectra data verified that our photosensitizers were independent of tissue-endogenous O₂ in PDT.

In addition, the tumor model was confirmed at a hypoxic state by immunofluorescence staining based on a hypoxia-associated protein, hypoxia-inducible factor 1- α (HIF-1 α) (Fig. 5c in the main text of our revised manuscripts). Besides, tumor blood vessels were further labelled using an anti-CD31 antibody (Fig. 5c in the main text of our manuscripts). As is well known, oxygen is mainly transported by blood vessels in vivo. However, only a limited number of tumor blood vessels (Red) were found, which also demonstrated tumor hypoxia even though we used the volume of tumors around 60 mm³.

To further address the Reviewer's concerns more effectively, the PDT experiments were carried out when the larger volume of tumors was around 500 mm³ (Figure R2a). In vivo PDT evaluation of BOH NPs with large-size 4T1 tumors (500 mm³) also confirmed the excellent anticancer performance, which further demonstrated the effectiveness of the developed PSs. As shown in Fig. R2b, the tumor model we established was confirmed at a severe hypoxic state by immunofluorescence staining based on a hypoxia-associated protein HIF-1 α . Besides, tumor blood vessels were further labelled using an anti-CD31 antibody. As is well known, oxygen is mainly transported by blood vessels in vivo. However, scarcely any tumor blood vessels (Red) were found, which also demonstrated tumor hypoxia. Next, the phototoxicity of BOH NPs for tumor in vivo was further evaluated in 4T1 tumor-bearing mice. 4T1-tumor-bearing female BALB/c mice with a tumor volume of 500 mm³ were randomly divided into four groups (n = 5), including (i) PBS group, (ii) BOH NPs group, the 4T1 tumor-bearing mice were only intravenously injected with BOH NPs (1 mg mL⁻¹, 200 μ L) without irradiation; (iii) 808 nm laser group, the mice were only irradiated under 808 nm laser (15 mW cm⁻²) for 10 min; (iv) BOH NPs + 808 nm laser group, the 4T1 tumor-bearing mice was intravenously injected with BOH NPs (1 mg mL⁻¹, 200 μ L), then with 808 nm laser (15 mW cm⁻²) irradiation at 12 h post-injection. The body weight of the mice in each group showed no significant change during the 12-day treatment period (Figure R2c), suggesting that PDT process exhibited negligible adverse effects on the health of mice. The change in tumor volume was monitored to evaluate the therapeutic effects. As shown in Figure R2d, the tumor volume of 808 nm laser group and BOH NPs group increased remarkably until about 1600 mm³. In contrast, the tumor volume in the BOH NPs + 808 nm laser group gradually decreased and was finally maintained at about 150 mm³. The representative tumor images (Figure R2e) and average tumor weight (Figure

R2f) further demonstrated the excellent antitumor efficacy of BOH NPs with light irradiation. The Hematoxylin and eosin (H&E) staining and terminal deoxynucleotidyl transferase dUTP nick end labelling (TUNEL) immunofluorescence staining assay further verified remarkable antitumor efficacy of BOH NPs with irradiation of 808 nm laser (Figure R2g).

The description has been supplemented in the main text (Page 26, Lines 499-501 and Page 27, Lines 502-516). The related data and detailed discussion have been added to the Supporting Information (Page S29, Lines 421-434 and Page S37, Lines 663-679).

Figure R2. In vivo PDT evaluation of BOH NPs with large-size tumors (500 mm³). (a) Schematic illustration of BOH NP-mediated PDT in a large volume of tumors (500 mm³). (b) Immunofluorescence imaging of tumor slices. The tumor blood vessels (red) are stained with the anti-CD31 antibody, hypoxia-related protein HIF- α is stained with the anti-HIF-1 α antibody (green). The (c) body weights and (d) tumor growth curves of the mice in vivo PDT study after intravenous injection of BOH NPs (1 mg mL⁻¹, 200 μ L). The tumors of BOH NPs and BOH NPs + 808 nm groups were irradiated by an 808 nm laser with a power density of 15 mW cm⁻² for 10 min. Mean \pm SD, n = 5. ***p < 0.001. (e) Photos of tumors collected from the mice in different groups at the end of PDT. (f) The mean weight of tumors separated from mice after different treatments. Mean \pm SD, n = 5. ***p < 0.001. (g) Representative H&E and TUNEL stained slices of tumors were collected from 4T1-tumor-bearing mice on the sixth day of the treatments in vivo PDT study.

Reviewer #2 (Remarks to the Author):

This manuscript (NCOMMS-23-37329) investigates new oxygen independent photosensitizers activated by ultralow-power near-infrared (NIR) excitation for photodynamic therapy. The manuscript is well organized and the experiments have been carefully designed and carried out. The idea and the aims of the work are interesting and the development of new photosensitizers deserves efforts and scientific investment. However, some relevant aspects are unclear or poorly discussed. Therefore, I recommend this manuscript for publication in Nature Communications, after some corrections. Some comments that, in my opinion, should be considered in a revised version:

Reply: We greatly appreciate the Reviewer for the highly positive and insightful comments guiding the revision of our manuscript. We will diligently address each of Reviewer's concerns in our revised version.

- The classification of mechanisms is wrong or, at least, in conflict with some reviews recently published that should be taken into account to establish the types of mechanism. Please, see the following review articles: Photochem. Photobiol. 93, 912–919 (2017); Photochem. Photobiol. 97, 1456–1483 (2021); Photochem. Photobiol. 99, 313–334 (2023).

Reply: We appreciate the Reviewer's suggestion. After carefully reading the reviews, we have incorporated them into the revised manuscript as references 17, 18 and 19.

We appreciate the Reviewer's perspective on the classification of mechanisms. To avoid confusion, we decided not to classify our PSs at this time. Our novel PSs generate $\bullet\text{O}_2^-$ or $\bullet\text{OH}$ through electron and hole transfer between the excited PSs and adjacent H_2O , eliminating the reliance on O_2 , which is different from both Type I and Type II.

Therefore, we deleted "type-III" in the revised manuscript.

- Superoxide radical anion and hydroxyl radical are not efficient species for PDT. Again, revise the articles mentioned in the previous paragraph and references cited therein and hypothesize why the photosensitizers works well although they produce species with low reactivity (superoxide) or with very low lifetime (hydroxyl radical).

Reply: Thank you very much for Reviewer's question. The effectiveness of the designed polymer photosensitizer is attributed to its remarkable ability to generate high levels of reactive oxygen species (ROS), surpassing the ROS production of the clinically approved photosensitizer methylene blue (MB) by more than 50-fold under equivalent hypoxic conditions. In addition, unlike conventional type-I and type-II photosensitizers, such as MB, the therapy effects severely depend on the limited oxygen available within the tumor for ROS generation, our probe facilitates the conversion of the abundant and ubiquitous water present in tissues into ROS through electron and hole transfer processes. Moreover, the designed polymer photosensitizer implemented deep-tissue-permeation 808 nm photoexcitation, which further improves the outcome of therapy. These enhancements collectively contributed to the success of our therapeutic strategy, overcoming the limitations associated with the low reactivity (superoxide) or short lifetime (hydroxyl radical) of the generated species.

- It would seem that the photosensitizer might react directly with biological substrates, for example, oxidizable biomolecules like nucleobases or amino acid residues. These pathways might contribute to the photodamage. This point is not discussed in detail and I think it is worth considering it.

Reply: We deeply appreciate the insightful question posed by the Reviewer. The prospect of direct reactions between the photosensitizer and biological substrates, such as oxidizable biomolecules, indeed warrants consideration. In response, we wish to elaborate on this aspect with careful consideration of both reaction thermodynamic and dynamic factors, as outlined in our manuscript.

The envisioned scenario where the photosensitizer directly interacts with biomolecules relies on two fundamental premises. Firstly, the redox potential between the photosensitizer and biological substrates must align favorably, providing a thermodynamic foundation for the reaction to occur. Secondly, the proximity between the photosensitizer and biological substrates is a crucial dynamic factor determining the reaction rate. Effective contact necessitates sufficient closeness between the photosensitizer and biomolecules. In the context of our proposed mechanism, it's vital to highlight that the limited propagation distance of electron-hole pairs in water imposes a constraint on the spatial range for reactions. In addition, the large size of biomolecules could block penetration and reactivity within the nanoscale particles of the photosensitizers. Therefore, electron-hole pairs of photosensitizers react preferentially with water from a competitive point of view.

To further address the Reviewer's concerns regarding potential photodamage, we conducted thorough investigations into physiological parameters in mice after *in vivo* treatments (A detailed PDT experiment method is described in Figure R2). Hematological data analysis (Figure R3), blood biochemical assays (Figure R4), and histological examinations of vital organs (Figure R5) collectively indicate that the treatment does not induce discernible photodamage. These comprehensive analyses provide additional assurance regarding the safety profile of our therapeutic approach.

The description has been supplemented in the main text (Page 27, Lines 517-523; Page 29, Lines 561-567; Page 30, Lines 568-569). The related data and detailed discussion have been added to Supporting Information (Page S30, Lines 437-459 and Page S31, Lines 460-470).

Figure R3. Haematological data analysis after the treatment sessions ended.

The haematological data include white blood cells (WBC), red blood cells (RBC), platelets (PLT), hemoglobin (HGB), hematocrit (HCT), mean corpuscular volume (MCV), mean corpuscular hemoglobin (MCH), and mean corpuscular hemoglobin concentration (MCHC). Compared with the control group, all the parameters in the BOH-NPs + 808 nm laser-treated groups did not show any significant differences, indicating that the therapy didn't cause obvious infection and inflammation in the treated mice.

Figure R4. Blood biochemical assay after the treatment sessions ended.

Blood biochemical analysis, including Hepatic functional indexes (ALT, AST, TBA, ALKP and GGT) and renal function indexes (UA, UREA and CREA), were examined. No meaningful difference can be observed between the BOH-NPs + 808 nm laser-treated groups and the PBS control group, suggesting that the blood chemistry of mice is not affected by treatment. Furthermore, hepatic and renal functional indexes, such as ALT, AST and CREA, demonstrate that the therapy induces no significant adverse effects on basic liver and kidney in mice.

Figure R5. H&E data (haematoxylin and eosin-stained images) obtained from the liver, spleen, kidney, heart and lung of the BOH-NPs-treated mice at 30 day post-injection.

H&E images of tissues (heart, liver, spleen, lung and kidney) harvested from PBS control mice and BOH-NPs + 808 nm laser-treated mice after the treatment sessions ended. The mice treated with buffered saline were used as blank controls (n = 3). No noticeable signal of organ damage can be observed after treatment from the two groups, suggesting no apparent histological abnormalities or lesions after the treatment.

Reviewer #3 (Remarks to the Author):

The study by Tang et. al. demonstrated an oxygen-independent PDT agent based on developing novel semiconducting polymers by rational skeleton design and precise side-chain engineering. It was verified that 808 nm photoexcitation can sensitize H₂O into •O₂⁻ and •OH under oxygen-free conditions. The investigation of photooxidation mechanisms with 18O-labelled mass spectra is interesting and convincing. In the biological tests, this system showed satisfying anticancer effect under a low-power laser irradiation and good selectivity of tumor imaging by responding to H₂O₂. However, improvements can be seen if the author can address the following issues:

Reply: We greatly appreciate the invaluable feedback from the Reviewer. Reviewer's insights are highly regarded, and we appreciate the opportunity to enhance our work based on Reviewer's valuable feedback.

Major concerns:

1. It was well-known that photosensitizers should be activated to its singlet states and triplet states for sensitizing oxygen or other substrates. It should be necessary to characterize the excited states of PSs by both theoretical methods and experimental methods, for example, transient spectrum study, energy calculations, and lifetime measurements of the singlet/triplet states. Also, it is exciting that the reported material can work under low-power 808 nm laser, but the reasons remain unclear. The authors are suggested to provide absorption coefficient, quantum yield of ROS or any other helpful information to further showcase the advantages of this PS as compared to the existing NIR PSs.

Reply: Thank you for your insightful queries. A point-by-point reply is included below.

Comment 1: It was well-known that photosensitizers should be activated to its singlet states and triplet states for sensitizing oxygen or other substrates. It should be necessary to characterize the excited states of PSs by both theoretical methods and experimental methods, for example, transient spectrum study, energy calculations, and lifetime measurements of the singlet/triplet states.

Reply: We highly value the insightful comments provided by the reviewer. We have undertaken meticulous computational simulations. Considering that the addition of the side chain will increase the calculation difficulty, we only calculated the skeleton structure with 4 units. The side chains of NTalk and NTgly are the same, so we calculated the skeleton structure of NTOalk and NTalk/NTgly with three units as a model (Figure R6a and e). The data is presented in the attached Figure R6, revealing noteworthy insights.

We carried out time-dependent density functional theory (TD-DFT) calculations of the singlet and the triplet excited states of NTOalk (Figure R6b-d) and NTalk/NTgly (Figure R6f-h). According to perturbation theory, a smaller singlet–triplet energy gap (ΔE_{ST}) and greater spin–orbit coupling (SOC) constants facilitate higher Φ_{ISC} . On the basis of simple energetic arguments, the T_n states with small ΔE_{ST} (± 0.3 eV) found in NTOalk and NTalk/NTgly model (Figure R6) resulted in a high intersystem crossing process (ISC) rate. By contrast, significant progress is particularly evident with the fact that the $S_1 \rightarrow T_4$ ΔE_{ST} (0.0209 eV) of NTOalk was significantly smaller than the $S_1 \rightarrow T_5$ ΔE_{ST} of NTalk/NTgly (0.0529 eV) (Figure 6b and 6f), which should facilitate the ISC process. In fact, NTOalk is a type-I PS. Diverging from the Type-I mechanism, which hinges on facilitating ISC to extend electron excited state lifetimes to boost collision probabilities with O_2 , the generation mechanism of ROS by NTalk/NTgly is not solely dependent on the ISC mechanism. Their rational design of the PS skeleton allows water to scavenge holes, thereby reducing electron-hole recombination efficiently. Furthermore, the incorporation of glycol side chains, in contrast to conventional alkyl chains, enhances the permittivity of the PS to impede rapid electron-hole recombination. The synergy of these design elements encourages a larger participation of electrons in reactions with O_2 , contributing to an overall improvement in reaction efficiency.

The 808 nm absorption and NIR-II fluorescence (1000-1700 nm) of our PSs have posed challenges in finding a suitable apparatus for testing the transient and lifetime characteristics of the excited states. Despite our efforts in reaching out to various research groups, we are very sorry that we were not able to find available equipment. Concerning the experimental aspect, we remain committed to exploring the excited states of this material once an instrument becomes available. We earnestly request your

forgiveness as we are currently unable to provide this data.

Figure R6. Chemical structures of the model of (a) NTOalk and (e) NTalk/NTgly. Singlet and triplet energy levels of different model of (b) NTOalk and (f) NTalk and NTgly. Calculated energy levels and possible ISC channels of different model of (c,d) NTOalk and (g,h) NTalk and NTgly. S and T stand for singlet, and triplet, respectively. The arrows refer to the major ISC channels, respectively. TD-DFT study of the CBT-dominant ISC. Schematic diagrams showing the computed energy levels and the probable ISC channels from the S_1 state to higher- or lower-lying triplet states (T_n). The triplet states in green boxes are those levels with energy higher than $E_{S_1} + 0.3$ eV or lower than $E_{S_1} - 0.3$ eV. The triplet states labelled in green contain the same transition configuration compositions as the S_1 state. Average spin-orbit

coupling (SOC) constants between S_1 and the involved triplet states (the larger SOC, the higher the ISC possibility) are also shown. Calculations were carried out with the Gaussian 09 package[2] and ORCA program.[3] S0 geometry optimizations of photosensitizers were performed with the density functional theory (DFT)/m062x in the gas phase. The 6-31g(d) basis set was used for all the atoms. Frequency calculations at the same level of theory were performed. Spin-orbital coupling matrix elements were investigated by ORCA program and time dependent density functional theory (TD-DFT). References: [1] Gaussian 09, Revision D.01, Frisch, M. J. et. al., Wallingford CT, 2013. [2] F. Neese, WIREs Comput. Mol. Sci. 2012, 2, 73-78.

Comment 2: Also, it is exciting that the reported material can work under low-power 808 nm laser, but the reasons remain unclear.

Reply: We highly appreciate the valuable comments from the Reviewer. High ROS yield ensures the attainment of desired therapeutic outcomes even at low light intensities. The high ROS yield is primarily attributed to the rational PS skeleton design and precise side-chain engineering. This design minimizes the rapid recombination of photon-generated electron-hole pairs, ensuring the supply of free electrons and holes for the H_2O redox reaction. The rational PS skeleton design enables the oxidation of H_2O into O_2 and H^+ under 808 nm laser irradiation, with subsequent conversion into $\bullet O_2^-$ or $\bullet OH$ by in-situ electrons. Water plays a dual role, providing O_2 and H^+ while scavenging holes to reduce electron-hole recombination significantly. This facilitates more electrons participating in reactions with O_2 and H^+ , enhancing overall reaction efficiency. Unlike the Type-I mechanism based on increasing electron triplet state lifetimes by facilitating the intersystem crossing process, our approach focuses on scavenging holes, which results in a more effective reduction of rapid electron-hole recombination, improving reaction efficiency between O_2 and electrons. Additionally, hydrophilic glycol side chains, instead of typical alkyl chains, increase PS permittivity, also inhibiting rapid electron-hole recombination and simultaneously ensuring sufficient H_2O molecules around electron-hole pairs produced by hydrophobic PS skeleton. The combined optimizations in thermodynamic and dynamic factors, including suitable redox potentials and glycol side chain modifications, synergistically contribute to the elevated $\bullet O_2^-$ and $\bullet OH$ production efficiency of our photosensitizers, which is over 50-fold higher than clinically approved MB under identical conditions. Consequently, our photosensitizers can achieve high-level ROS production even under low-power 808 nm laser, ensuring effective therapeutic outcomes.

Comment 3: The authors are suggested to provide absorption coefficient, quantum yield of ROS or any other helpful information to further showcase the advantages of this PS as compared to the existing NIR PSs.

Reply: We thank the Reviewer for bringing up this important point. We recognize the significance of the quantum yield in assessing ROS generation, especially for singlet oxygen. Unlike singlet oxygen, there is no widely accepted formula to compute quantum yields of superoxide and hydroxyl radicals. In response to feedback, we've analyzed crucial parameters for comparing our photosensitizer with NIR counterparts.

To provide a robust framework for evaluation, we have compiled a detailed table (Table 1) featuring probe names, excitation wavelength, power, irradiation time, and the fold increase in ROS probe fluorescence. It is important to note that we solely provide a summary of the reactive oxygen species (ROS) production capabilities of photosensitizers when activated by NIR light. As shown in Table 1, some of their photosensitizers show relatively high ROS production efficiency **under normoxic conditions** in vitro. However, we also note that these studies (J.Am.Chem.Soc.2019, 141, 16243; J.Am.Chem.Soc. 2020, 142, 11, 5380 and J.Am.Chem.Soc. 2018, 140, 44, 14851) reported that the cell death ratios of PDT obviously decrease in hypoxia conditions due to the limitation of their O_2 -

dependent type-I ROS production mechanism, not to mention highly hypoxic tumors in vivo. By contrast, our O₂-independent NTgly PS shows a high ROS production rate whether under normal oxygen or hypoxic conditions, which is very beneficial for hypoxic tumor therapy in vivo.

Table 1: Summary of recent NIR-activated organic photodynamic theranostic materials

Probe Name/ References	Excitation Wavelength	Type	Power/ Irradiation Time	Normoxia conditions	
				Fluorescence Increase of •O ₂ ⁻ Probe	Fluorescence Increase of •OH Probe
NTgly NPs (This work)	808 nm	O ₂ -independent new mechanism	15 mW cm ⁻² 5 min	38-fold	36.33-fold
Methylene blue (MB) (This work)	635 nm	O ₂ -dependent Type-I	15 mW cm ⁻² 5 min	2.53-fold	1.52-fold
ENBOS (J. Am. Chem. Soc. 2019, 141, 2695)	660 nm	O ₂ -dependent Type-I	2 mW cm ⁻² 6 min	4.8-fold	0-fold
F8CA NPs (ACS Macro Lett. 2023, 12, 10, 1365)	660 nm	O ₂ -dependent Type-I	1000 mW cm ⁻² 6 min	9-fold	10-fold
DPIC NPs (Sci. China: Chem., 2022, 65, 1134)	808 nm	O ₂ -dependent Type-I	1000 mW cm ⁻² 10 min	/	2-fold
DPTTIC NPs (Chem. Commun., 2022, 58, 10353)	980 nm	O ₂ -dependent Type-I	1000 mW cm ⁻² 10 min	/	~8-fold
TQ-TPA NMs (Mater. Chem. Front., 2023, 7, 3657)	660 nm	O ₂ -dependent Type-I	300 mW cm ⁻² 6 min	/	9.5-fold
TPAPEVDPP NPs (ACS Nano 2022, 16, 4162)	660 nm	O ₂ -dependent Type-I	500 mW cm ⁻² 1.5 min	40-fold	1.1-fold
Cyl (ACS Mater. Lett. 2023, 5, 11, 3058)	660 nm	O ₂ -dependent Type-I	30 mW cm ⁻² 10 min	4.5-fold	/
TPA-BTZ@PEG2000 NPs (Biomater. Sci., 2022, 10, 4785)	635 nm	O ₂ -dependent Type-I	400 mW cm ⁻² 9 min	9-fold	1.1
BMIC-BO-4CINPs (Angew. Chem.Int. Ed. 2023, 62, e2023 03476)	880 nm	O ₂ -dependent Type-I	300 mW cm ⁻² 3 min	45-fold	40-fold
CyBr (Adv. Mater. 2023. DOI:10.1002/adma.202305243)	660 nm	O ₂ -dependent Type-I	50 mW cm ⁻² 10 min	3.7-fold	/
DCTBT NPs (Biomaterials)	808 nm	O ₂ -dependent Type-I	800 mW cm ⁻² 6 min	100-fold	45-fold

2022,283, 121476)					
CMTP1 (Anal. Chem. 2021, 93, 35, 12059)	808 nm	O ₂ -dependent Type-I	1000 mW cm ⁻² 5 min	3.5-fold	/
CMTP2 (Anal. Chem. 2021, 93, 35, 12059)	808 nm	O ₂ -dependent Type-I	1000 mW cm ⁻² 5 min	10-fold	/
DTTVBI (J. Am. Chem.Soc.2023,14 5,334)	808 nm	O ₂ -dependent Type-I	800 mW cm ⁻² 6 min	12-fold	~ 0-fold

2. In Fig. 1g and Fig. S19, it was demonstrated that the NTgly showed a much larger red shift after self-assembly. Compared to NTalk (redshifted 50 nm) and NTOalk (redshifted 40 nm), NTgly showed a much larger red shift (150 nm). Thus, the reason may not only be aggregation-induced red shift. The authors should provide more information to explain this phenomenon.

Reply: We highly appreciate the valuable comments from the Reviewer. Among the three polymers of NTOalk, NTalk, and NTgly, the NTalk and NTgly possess the same conjugated backbone and similar molecule weight but have different side chains. NTalk integrates an alkyl chain, whereas NTgly incorporates a glycol side chain of elevated polarity, heightened hydrophilicity, and increased flexibility. Therefore, we use NTalk and NTgly to explain the large red shift phenomenon of NTgly. NTalk and NTgly in dichloromethane have absorption maxima at 660 and 595 nm, respectively (Supplementary Fig. 19 in the revised Supporting Information). In contrast, the NTalk NPs and NTgly NPs exhibit absorption peaks at 710 and 745 nm, respectively (Fig. 1g in the revised main text). The large redshift of NTgly is mainly attributed to the fact that the maximum absorption of NTgly undergoes a blue shift of 65 nm in solution compared to the NTalk, whereas in the nanoparticle state, it experiences a redshift of 35 nm. According to previous reports (*Chem. Mater.* 2018, 30, 2945.; *Macromolecules* 2023, 56, 2092. and *ACS Nano* 2022, 16, 21303.), the blue shift of the maximum absorption in the solution state and the red shift in the nanoparticles primarily arises from less aggregation in solution and enhanced intermolecular packing in nanoparticles of polymers with glycol side chains compared to those with alkyl side chains.

Therefore, we deleted the sentence "which attributes to stronger aggregation of polymer PS skeletons in water." in the revised manuscript.

3. There is a lack of biosafety testing. The dark toxicity of the NPs should be studied. Besides, after in vivo treatments, some physiological parameters of mice should be investigated.

Reply: A point-by-point reply is included below.

Comment 1: There is a lack of biosafety testing. The dark toxicity of the NPs should be studied.

Reply: We deeply appreciate the invaluable feedback provided by the Reviewer. To further investigate the dark toxicity of BOH-NPs, we used healthy C57Bl/6 mice (7 weeks old) to investigate the change in mice weight, hematology indicators, blood biochemical analysis, hepatic and renal functional indexes and haematoxylin and eosin staining analysis post-injection. Firstly, 6 mice were randomly divided into 2 groups (n = 3 per group) and subjected to variable conditions, including (1) a control group without any treatment, (2) BOH-NPs (dose ~ 100 mg/kg) intravenously injected into the mice. Their body weight was monitored post-injection to assess healthy condition changes.

The 36 mice were randomly divided into 2 groups ($n = 18$ per group) and subjected to variable conditions, including (1) a control group without any treatment and (2) BOH-NPs (dose ~ 100 mg/kg) intravenously injected into the mice. The haematological, blood biochemical and histological analyses were performed at time points of 1, 7 and 30 days post-injection, respectively. The mice were then sacrificed. The heart, lung, liver, spleen and kidney were embedded in paraffin, sectioned, and stained with hematoxylin and eosin (H&E), respectively. The tissues were subsequently processed for histopathological examination under a light microscope.

The description has been supplemented in the main text (Page 28, Lines 525-542). The related data and detailed discussion have been added to Supporting Information (Page S31, Lines 478-480; Page S32, Lines 481-506; Page S33, Lines 509-526; Page S34, Lines 527-534; Page S37, Lines 681-699 and Page S38, Line 700).

(b)

Behaviours and appearance	Eating			Drinking			Hair Colour			Glossiness		
	1 day	7 day	30 day	1 day	7 day	30 day	1 day	7 day	30 day	1 day	7 day	30 day
Control	No any significant differences			No any significant differences			No any significant differences			No any significant differences		
BOH-NPs	No any significant differences			No any significant differences			No any significant differences			No any significant differences		

Figure R8. (a) Body weight curve and (b) behaviours and appearance of BOH-NPs (100 mg kg^{-1}) treated mice. Body weight changes of healthy mice with and without intravenous injections of BOH-NPs ($n = 3$ mice in each group) over 30 days. Data were presented as mean \pm s.d.

To assess the influence of BOH-NPs on the development and growth of mice, body weight was continuously recorded. Each mouse was injected with $100 \mu\text{L}$ of buffered saline (control) or buffered probe BOH-NPs (dosage: 100 mg kg^{-1}) by tail vein. Following these injections, the mice were weighed at different time points from 0 to 30 days. As shown in Figure R8, the mice injected with BOH-NPs showed no body weight loss compared with control mice without any treatment over the 30 days. Additionally, the mice did not show any significant differences in behaviors and appearance, including eating, drinking, hair color and glossiness, between the BOH-NPs-treated mice and the control groups. These results demonstrated that BOH-NPs didn't cause overall side effects in the mice.

Figure R9. Haematological data analysis of the mice intravenously injected with the BOH-NPs at 1, 7 and 30 days post-injection.

The data include white blood cells (WBC), red blood cells (RBC), platelets (PLT), hemoglobin (HGB), hematocrit (HCT), mean corpuscular volume (MCV), mean corpuscular hemoglobin (MCH), and mean corpuscular hemoglobin concentration (MCHC). Compared with the control group, all the parameters in the BOH-NPs-treated groups at all-time points did not show any significant differences, indicating that the BOH-NPs didn't cause obvious infection and inflammation in the treated mice.

Figure R10. Blood biochemical assay of BOH-NPs.

Hepatic functional indexes (alanine transaminase (ALT), aspartate transaminase (AST), total bile acid (TBA), alkaline phosphatase (ALKP) and gamma-glutamyl transferase (GGT)) and renal function indexes (uric acid (UA), urea nitrogen (UREA) and creatinine (CREA)) of the mice treated with and without BOH-NPs. Blood biochemical analysis, including Hepatic functional indexes (ALT, AST, TBA, ALKP and GGT) and renal function indexes (UA, UREA and CREA), were examined. No meaningful difference can be observed between the BOH-NPs-treated groups at all time points with the control group, suggesting that the blood chemistry of mice is not affected by BOH-NPs treatment. Furthermore, hepatic and renal functional indexes, such as ALT, AST and CREA, demonstrate that the BOH-NPs induce no significant adverse effects on basic liver and kidney in mice.

Figure R11. Histological data (haematoxylin and eosin-stained images) obtained from the liver, spleen, kidney, heart and lung of the BOH-NPs-treated mice at 1, 7 and 30 day post-injection. (Scale bar, 100 μm .)

Haematoxylin and eosin-stained images of tissues (heart, liver, spleen, lung and kidney) harvested from control mice and 1, 7 and 30 day after intravenous injection of BOH-NPs (100 mg kg^{-1}). The mice treated with buffered saline were used as blank controls ($n = 3$). No noticeable signal of organ damage can be observed during the whole treatment period from all the groups, suggesting no apparent histological abnormalities or lesions in the nanoprobe-treated groups for the test dose.

Comment 2: Besides, after in vivo treatments, some physiological parameters of mice should be investigated.

Reply: We highly appreciate the valuable comments from the Reviewer. After the therapy sessions ended (Detailed PDT experiment method are described in Figure R2), three mice in the PBS and BOH-NPs + 808 nm laser group were randomly selected to perform the haematological, blood biochemical and histological analyses, respectively. The mice were then sacrificed. The heart, lung, liver, spleen and kidney were embedded in paraffin, sectioned, and stained with hematoxilin and eosin (H&E), respectively. The tissues were subsequently processed for histopathological examination under a light microscope.

The description has been supplemented in the main text (Page 27, Lines 517-523). The related data and detailed discussion have been added to the Supporting Information (Page S30, Lines 437-459; Page S31, Lines 460-470).

Figure R12. Haematological data analysis after the treatment sessions ended.

The haematological data include white blood cells (WBC), red blood cells (RBC), platelets (PLT), hemoglobin (HGB), hematocrit (HCT), mean corpuscular volume (MCV), mean corpuscular hemoglobin (MCH), and mean corpuscular hemoglobin concentration (MCHC). Compared with the control group, all the parameters in the BOH-NPs + 808 nm laser-treated groups did not show any significant differences, indicating that the therapy didn't cause obvious infection and inflammation in the treated mice.

Figure R13. Blood biochemical assay after the treatment sessions ended.

Blood biochemical analysis, including Hepatic functional indexes (ALT, AST, TBA, ALKP and GGT) and renal function indexes (UA, UREA and CREA), were examined. No meaningful difference can be observed between the BOH-NPs + 808 nm laser-treated groups and the PBS control group, suggesting that the blood chemistry of mice is not affected by treatment. Furthermore, hepatic and renal functional indexes, such as ALT, AST and CREA, demonstrate that the therapy induces no significant adverse effects on basic liver and kidney in mice.

Figure R14. H&E data (haematoxylin and eosin stained images) obtained from the liver, spleen, kidney, heart and lung of the BOH-NPs-treated mice at 30 day post-injection. (Scale bar, 100 μm.)

H&E images of tissues (heart, liver, spleen, lung and kidney) harvested from PBS control mice and BOH-NPs + 808 nm laser-treated mice after the treatment sessions ended. The mice treated with buffered saline were used as blank controls (n = 3). No noticeable signal of organ damage can be observed after treatment from the two groups, suggesting no apparent histological abnormalities or lesions after the treatment.

Minor points:

1. In Fig 5a, BOH NPs have less liver accumulation than BOH NPs + NAC, why?

Reply: Thanks for the insightful comments. The imaging was conducted with mice in a prone position, exposing their dorsal side. In this configuration, only a portion of the liver fluorescence could penetrate through the back and be detected. It is possible that the spine or sternum obstructed some of the liver fluorescence, resulting in a different fluorescence signal. However, the ex vivo imaging of the excised organs showed a rational liver fluorescence signal between BOH NPs and BOH-NPs + NAC group (Supplementary Fig. 38 in the revised supplementary information).

2. In Fig. 5c, it is necessary to show the CD31 and HIF levels for the control groups.

Reply: Thanks for the constructive feedback. We used normal tissue from adjacent tumor margins as our control. As shown in Figure R15, the normal tissues exhibited more prominent vasculature (depicted in red) and less presence of HIF-1α (depicted in green) compared to the tumor tissues (Fig. 5c in the main text of our manuscripts), indicating that the tumor is in a hypoxic state.

The description has been supplemented in the main text (Page 25, Lines 470-473). The related data and detailed discussion have been added to the Supporting Information (Page S27, Lines 402-405).

Figure R15. Immunofluorescence imaging of tumor slices. The tumor blood vessels (red) are stained with the anti-CD31 antibody, hypoxia-related protein HIF-α is stained with the anti-HIF-1α antibody (green).

3. Balb/c should be revised as BALB/c.

Reply: We sincerely apologize for the mistakes. We have revised "Balb/c" into "BALB/c".

4. In Line 576, the author mentioned "the molal concentration of MB and NTgly is equal". How to determine the molal concentration of NTgly?

Reply: We appreciate the opportunity to clarify this. We calculated the molar concentration of the NTgly based on the average molecular weight measured by Gel Permeation Chromatography. Then, we use the mass of NTgly divided by its molecular weight.

Thank you in advance for your time and we look forward to hearing from you in due course.

Sincerely,

Prof. Bin Liu
Department of Chemical and Biomolecular Engineering
National University of Singapore
Singapore, 117585
Tel: 65-6516-8409
Fax: 65-6778-1936

Reviewers' Comments:

Reviewer #1:

Remarks to the Author:

The modified manuscript (NCOMMS-23-37329A) submitted by Tang et al. introduces a design strategy for creating oxygen-independent photosensitizers tailored for tumor PDT under ultralow-power NIR photoexcitation. While most of the concerns I raised have been effectively addressed, certain essential points still require clarification and additional supporting information to substantiate the title.

1. In this article, the authors underscore the oxygen-independent functionality of the photosensitizers, attributing it to the water oxidation by holes generated through NIR light, followed by oxygen reduction via electrons. However, comprehending the notion that a significant portion of the generated electrons remains "in situ" to facilitate the reduction of the newly formed O₂ from water oxidation is challenging, given that electron migration rates typically surpass those of holes. Additionally, the net ionic equations outlining the proposed mechanism introduce external O₂ as a crucial reactant (oxidation part: $2\text{H}_2\text{O} + 4\text{h}^+ \rightarrow \text{O}_2 + 4\text{H}^+$; reduction part: $\text{O}_2 + \text{e}^- \rightarrow \bullet\text{O}_2^-$ or $\text{O}_2 + 2\text{H}^+ + 2\text{e}^- \rightarrow 2\bullet\text{OH}$; net ionic equation1: $2\text{H}_2\text{O} + 3\text{O}_2 \rightarrow 4\bullet\text{O}_2^- + 4\text{H}^+$; net ionic equation2: $2\text{H}_2\text{O} + \text{O}_2 \rightarrow 4\bullet\text{OH}$). Although isotope labeling experiments support the possibility of ROS generation through the oxygen obtained by the oxidation of water, reliance on this pathway as the primary photosensitive mechanism is still uncertain, as mass spectrometry-based experiments offer only qualitative evidence. To substantiate the claim that the compounds operate in an oxygen-independent manner, a series of ROS detection tests without an O₂ atmosphere is essential.

2. Page 15 line 290, TMPO should be changed as DMPO.

Reviewer #2:

Remarks to the Author:

The authors have made an effort to improve the manuscript. I think that the main comments and criticisms have been addressed in a satisfactory manner. I recommend the publication of the manuscript in its present form.

Reviewer #3:

Remarks to the Author:

The authors have provided additional data and adequately responded the comments. The acceptance of this manuscript for publication is suggested.

January 14, 2024

Dear Reviewers

We thank the reviewers for their constructive comments, which have helped us improve our manuscript. Along with this letter, please find the revised electronic version of the manuscript (NCOMMS-23-37329B) entitled "Oxygen-Independent Organic Photosensitizer with Ultralow-Power NIR Photoexcitation for Tumor-Specific Photodynamic Therapy". Below we give a point-by-point response in blue text to the questions raised. All the changes are marked in red color in the revised manuscript for your reference. Thank you very much for your kind attention.

Our responses to the Reviewer's comments are summarized as follows:

COMMENTS TO AUTHOR:

Reviewer #1 (Remarks to the Author):

The modified manuscript (NCOMMS-23-37329A) submitted by Tang et al. introduces a design strategy for creating oxygen-independent photosensitizers tailored for tumor PDT under ultralow-power NIR photoexcitation. While most of the concerns I raised have been effectively addressed, certain essential points still require clarification and additional supporting information to substantiate the title.

Reply: We greatly appreciate the Reviewer for acknowledging the revised manuscript. We take the Reviewer's concerns seriously and have performed new experiments to support our claims.

1. In this article, the authors underscore the oxygen-independent functionality of the photosensitizers, attributing it to the water oxidation by holes generated through NIR light, followed by oxygen reduction via electrons. However, comprehending the notion that a significant portion of the generated electrons remains "in situ" to facilitate the reduction of the newly formed O₂ from water oxidation is challenging, given that electron migration rates typically surpass those of holes. Additionally, the net ionic equations outlining the proposed mechanism introduce external O₂ as a crucial reactant (oxidation part: $2\text{H}_2\text{O} + 4\text{h}^+ \rightarrow \text{O}_2 + 4\text{H}^+$; reduction part: $\text{O}_2 + \text{e}^- \rightarrow \bullet\text{O}_2^-$ or $\text{O}_2 + 2\text{H}^+ + 2\text{e}^- \rightarrow 2\bullet\text{OH}$; net ionic equation1: $2\text{H}_2\text{O} + 3\text{O}_2 \rightarrow 4\bullet\text{O}_2^- + 4\text{H}^+$; net ionic equation2: $2\text{H}_2\text{O} + \text{O}_2 \rightarrow 4\bullet\text{OH}$). Although isotope labeling experiments support the possibility of ROS generation through the oxygen obtained by the oxidation of water, reliance on this pathway as the primary photosensitive mechanism is still uncertain, as mass spectrometry-based experiments offer only qualitative evidence. To substantiate the claim that the compounds operate in an oxygen-independent manner, a series of ROS detection tests without an O₂ atmosphere is essential.

Reply: We appreciate the Reviewer's insightful queries. A point-by-point reply is included below.

Comment 1: In this article, the authors underscore the oxygen-independent functionality of the photosensitizers, attributing it to the water oxidation by holes generated through NIR light, followed by oxygen reduction via electrons. However, comprehending the notion that a significant portion of the generated electrons remains "in situ" to facilitate the reduction of the newly formed O₂ from water oxidation is challenging, given that electron migration rates typically surpass those of holes.

Reply: We thank the Reviewer for the valuable comments. We agree that electron migration rates could

surpass those of holes. To avoid confusion, we have deleted "in situ" in our revised manuscript.

Comment 2: Additionally, the net ionic equations outlining the proposed mechanism introduce external O₂ as a crucial reactant (oxidation part: $2\text{H}_2\text{O} + 4\text{h}^+ \rightarrow \text{O}_2 + 4\text{H}^+$; reduction part: $\text{O}_2 + \text{e}^- \rightarrow \bullet\text{O}_2^-$ or $\text{O}_2 + 2\text{H}^+ + 2\text{e}^- \rightarrow 2\bullet\text{OH}$; net ionic equation1: $2\text{H}_2\text{O} + 3\text{O}_2 \rightarrow 4\bullet\text{O}_2^- + 4\text{H}^+$; net ionic equation2: $2\text{H}_2\text{O} + \text{O}_2 \rightarrow 4\bullet\text{OH}$). Although isotope labeling experiments support the possibility of ROS generation through the oxygen obtained by the oxidation of water, reliance on this pathway as the primary photosensitive mechanism is still uncertain, as mass spectrometry-based experiments offer only qualitative evidence. To substantiate the claim that the compounds operate in an oxygen-independent manner, a series of ROS detection tests without an O₂ atmosphere is essential.

Reply: We sincerely thank the Reviewer's constructive feedback. In response to the Reviewer's comment, we have conducted additional ROS detection tests in the absence of oxygen. The oxygen-free environment was achieved by employing an AnaeroPack™-Anaero Anaerobic Gas Generator within a sealed transparent chamber (Figure R1a). The probe NTgly NPs produce a high level of ROS in the absence of O₂ (Figure R1b), which is consistent with our previous results obtained using a nearly oxygen-free environment (< 0.1 ppm) created in a glove box (Figure 2a-d), emphasizing the oxygen-independent nature of our photosensitizer. The related data have been added to the Supporting Information (Page S21, Lines 332-336). The detailed description and discussion have been supplemented in the main text (Page 12, Line 253 and Page 13, Lines 254-255).

Please also take note that in our system, both ($\text{O}_2 + \text{e}^- \rightarrow \bullet\text{O}_2^-$) and ($\text{O}_2 + 2\text{H}^+ + 2\text{e}^- \rightarrow 2\bullet\text{OH}$) are subsequent reactions of ($2\text{H}_2\text{O} + 4\text{h}^+ \rightarrow \text{O}_2 + 4\text{H}^+$). Neither ($\text{O}_2 + \text{e}^- \rightarrow \bullet\text{O}_2^-$) nor ($\text{O}_2 + 2\text{H}^+ + 2\text{e}^- \rightarrow 2\bullet\text{OH}$) is a complete half reaction.

Figure R1. (a) The AnaeroPack™-Anaero Anaerobic Gas Generator within a sealed transparent chamber. (b) The fluorescence intensity changes of $\bullet\text{O}_2^-$ probe (DHR 123) at 525 nm and $\bullet\text{OH}$ probe (HPF) at 515 nm for NTgly NPs ($5 \mu\text{g mL}^{-1}$) under 808 nm irradiation at 15 mW cm^{-2} for 5 min.

2. Page 15 line 290, TMPO should be changed as DMPO.

Reply: We acknowledge the valuable feedback from the Reviewer and sincerely apologize for our oversight. We have revised "TMPO" into "DMPO".

Reviewer #2 (Remarks to the Author):

The authors have made an effort to improve the manuscript. I think that the main comments and criticisms have been addressed in a satisfactory manner. I recommend the publication of the manuscript in its present form.

Reply: We greatly appreciate the Reviewer's highly positive recommendation.

Reviewer #3 (Remarks to the Author):

The authors have provided additional data and adequately responded the comments. The acceptance of this manuscript for publication is suggested.

Reply: We greatly appreciate the Reviewer's highly positive suggestion.

Thank you in advance for your time and we look forward to hearing from you in due course.

Sincerely,

Prof. Bin Liu
Department of Chemical and Biomolecular Engineering
National University of Singapore
Singapore, 117585
Tel: 65-6516-8409
Fax: 65-6778-1936

Reviewers' Comments:

Reviewer #1:

Remarks to the Author:

The authors have made efforts to improve the manuscript. I have no other questions. The revised manuscript is acceptable.